# Statistical and Computational Guarantees of Kernel Max-Sliced Wasserstein Distances

**Jie Wang**[1,2]  **March Boedihardjo**[3]  **Yao Xie**[4]

## Abstract

Optimal transport has been very successful for various machine learning tasks; however, it is known to suffer from the curse of dimensionality. Hence, dimensionality reduction is desirable when applied to high-dimensional data with low-dimensional structures. The kernel max-sliced (KMS) Wasserstein distance is developed for this purpose by finding an optimal nonlinear mapping that reduces data into 1 dimension before computing the Wasserstein distance. However, its theoretical properties have not yet been fully developed. In this paper, we provide sharp finite-sample guarantees under milder technical assumptions compared with state-of-the-art for the KMS $p$-Wasserstein distance between two empirical distributions with $n$ samples for general $p \in [1, \infty)$. Algorithm-wise, we show that computing the KMS 2-Wasserstein distance is NP-hard, and then we further propose a semidefinite relaxation (SDR) formulation (which can be solved efficiently in polynomial time) and provide a relaxation gap for the obtained solution. We provide numerical examples to demonstrate the good performance of our scheme for high-dimensional two-sample testing.

## 1. Introduction

Optimal transport (OT) has achieved much success in various areas, such as generative modeling [27, 49, 59, 61], distributionally robust optimization [24, 25, 73], non-parametric testing [63, 71, 76, 78, 80], domain adapta-tion [2, 12–14, 77], etc. See [62] for comprehensive reviews on these topics. The sample complexity of the Wasserstein distance has been an essential building block for OT in statistical inference. It studies the relationship between a population distribution $\mu$ and its empirical distribution $\frac{1}{n} \sum_{i=1}^{n} \delta_{x_i}$ with $x_i \sim \mu$ in terms of the "Wasserstein distance". Unfortunately, the sample size $n$ needs to be exponentially large in data dimension to achieve an accurate enough estimation [22], called the *curse of dimensionality* issue.

To tackle the challenge of high dimensionality, it is meaningful to combine OT with projection operators in low-dimensional spaces. Researchers first attempted to study the sliced Wasserstein distance [10, 11, 17, 38, 39, 54, 56], which computes the average of the Wasserstein distance between two projected distributions using random one-dimensional projections. Since a single random projection contains little information to distinguish two high-dimensional distributions, computing the sliced Wasserstein distance requires a large number of linear projections. To address this issue, more recent literature considered the **M**ax-**S**liced (**MS**) Wasserstein distance that seeks the *optimal* projection direction such that the Wasserstein distance between projected distributions is maximized [16, 45, 47, 60, 74]. Later, Wang et al. [75] modified the MS Wasserstein distance by seeking an optimal *nonlinear* projection belonging to a ball of reproducing kernel Hilbert space (RKHS), which we call the **K**ernel **M**ax-**S**liced *(**KMS**) Wasserstein distance*. The motivation is that a nonlinear projector can be more flexible in capturing the differences between two high-dimensional distributions; it is worth noting that KMS Wasserstein reduces to MS Wasserstein when specifying a dot product kernel.

Despite promising applications of the KMS Wasserstein distance, its statistical and computational results have not yet been fully developed. From a statistical perspective, Wang et al. [75] built concentration properties of the empirical KMS Wasserstein distance for distributions that satisfy the projection Poincaré inequality and the Poincaré inequality [42], which could be difficult to verify in practice. From a computational perspective, the authors therein developed a gradient-based algorithm to approximately compute the empirical KMS Wasserstein distance. However, there is no

---

[1]School of Artificial Intelligence, The Chinese University of Hong Kong, Shenzhen, Shenzhen, China [2]School of Data Science, The Chinese University of Hong Kong, Shenzhen, Shenzhen, China [3]Department of Mathematics, Michigan State University, East Lansing, USA [4]School of Industrial and Systems Engineering, Georgia Institute of Technology, Atlanta, USA. Correspondence to: Yao Xie <yao.xie@isye.gatech.edu>.

*Proceedings of the 42nd International Conference on Machine Learning*, Vancouver, Canada. PMLR 267, 2025. Copyright 2025 by the author(s).

theoretical guarantee on the quality of the local optimum solution obtained. In numerical experiments, the quality of the solution obtained is highly sensitive to the initialization.

To address the aforementioned limitations, this paper provides new statistical and computational guarantees for the KMS Wasserstein distance. Our key contributions are summarized as follows.

- We provide a non-asymptotic estimate on the KMS $p$-Wasserstein distance between two empirical distributions based on $n$ samples, referred to as the *finite-sample guarantees*. Our result shows that when the samples are drawn from identical populations, the rate of convergence is $n^{-1/(2p)}$, which is dimension-free and optimal in the worst case scenario.

- We analyze the computation of KMS 2-Wasserstein distance between two empirical distributions based on $n$ samples. First, we show that computing this distance exactly is NP-hard. Consequently, we are prompted to propose a semidefinite relaxation (SDR) as an approximate heuristic with various guarantees.

  - We develop an efficient first-order method with biased gradient oracles to solve the SDR, the complexity of which for finding a $\delta$-optimal solution is $\widetilde{\mathcal{O}}\left(n^2\delta^{-2}\cdot\max(n,\delta^{-1})\right)$. In comparison, the complexity of the interior point method for solving SDR is $\widetilde{\mathcal{O}}(n^{6.5})$ [5].

  - We derive theoretical guarantees for the optimal solutions from the SDR. We show that there exists an optimal solution from SDR that is at most rank-$k$, where $k \triangleq 1 + \lfloor\sqrt{2n+9/4}-3/2\rfloor$, whereas computing the KMS distance exactly requires a rank-1 solution. We also provide a corresponding rank reduction algorithm designed to identify such low-rank solutions from the pool of optimal solutions of SDR.

- We exemplify our theoretical results in non-parametric two-sample testing, human activity detection, and generative modeling. Our numerical results showcase the stable performance and quick computational time of our SDR formulation, as well as the desired sample complexity rate of the empirical KMS Wasserstein distance.

**Literature Review**

In the following, we compare our work with the most closely related literature, and defer the detailed comparision of KMS Wasserstein distance with other variants of OT divergences in Appendix A.

The study on the statistical and computational results of MS and KMS Wasserstein distances is popular in the ex-

isting literature. From a statistical perspective, existing results on the rate of empirical MS/KMS Wasserstein are either dimension-dependent, suboptimal or require regularity assumptions (e.g., log-concavity, Poincaré inequality, projection Bernstein tail condition) on the population distributions [3, 47, 57, 74], except for the very recent literature [9] that provides a sharp, dimension-free rate for MS Wasserstein with data distributions supported on a compact subspace but without regularity assumptions. From the computational perspective, there are two main approaches to compute such distances. One is to apply gradient-based algorithms to find local optimal solutions or stationary points, see, e.g., [33–35, 46, 75]. Unfortunately, due to the highly non-convex nature of the optimization problem, the quality of the estimated solution is unstable and highly depends on the choice of initial guess. The other is to consider solving its SDR instead [60], yet theoretical guarantees on the solution from convex relaxation are missing. Inspired by existing reference [4, 18, 44, 58] that studied the rank bound of SDR for various applications, we adopt their proof techniques to provide similar guarantees for computing KMS in Theorem 4.5. Besides, all listed references add entropic regularization to the inner OT problem and solve the regularized version of MS/KMS Wasserstein distances instead, while the gap between the solutions from regularized and original problems could be non-negligible.

Our proposed KMS Wasserstein distance is also closely related to [83], which considers the Wasserstein distance between pushforward measures $\Phi_{\#}\mu$ and $\Phi_{\#}\nu$ via an implicitly defined kernel map $\Phi(\cdot)$. We will show that our metric can be viewed as the max-sliced version of this setting (Remark 2.6). A key distinction is that our method enjoys sharp statistical convergence rates, which are difficult to obtain in Zhang et al. [83] due to the curse of dimensionality of Wasserstein.

## 2. Background

We first introduce the definition of Wasserstein and KMS Wasserstein distances below.

**Definition 2.1** (Wasserstein Distance). Let $p \in [1,\infty)$. Given a normed space $(\mathcal{V}, \|\cdot\|)$, the $p$-Wasserstein distance between two probability measures $\mu, \nu$ on $\mathcal{V}$ is defined as

$$W_p(\mu,\nu) = \left(\min_{\pi\in\Gamma(\mu,\nu)}\int \|x-y\|^p\,\mathrm{d}\pi(x,y)\right)^{1/p},$$

where $\Gamma(\mu,\nu)$ denotes the set of all probability measures on $\mathcal{V}\times\mathcal{V}$ with marginal distributions $\mu$ and $\nu$.

**Definition 2.2** (RKHS). Consider a symmetric and positive definite kernel $K : \mathcal{B}\times\mathcal{B} \to \mathbb{R}$, where $\mathcal{B} \subseteq \mathbb{R}^d$. Given such a kernel, there exists a unique Hilbert space $\mathcal{H}$, called RKHS, associated with the reproducing kernel $K$. Denote by $K_x$ the kernel section at $x \in \mathcal{B}$ defined by $K_x(y) = K(x,y), \forall y \in$

$\mathcal{B}$. Any function $f \in \mathcal{H}$ satisfies the reproducing property $f(x) = \langle f, K_x \rangle_{\mathcal{H}}, \forall x \in \mathcal{B}$. For $x, y \in \mathcal{B}$, it holds that $K(x, y) = \langle K_x, K_y \rangle_{\mathcal{H}}$.

**Definition 2.3** (KMS Wasserstein Distance). Let $p \in [1, \infty)$. Given two distributions $\mu$ and $\nu$, define the $p$-KMS Wasserstein distance as

$$\mathcal{KMS}_p(\mu, \nu) = \max_{f \in \mathcal{H}, \, \|f\|_{\mathcal{H}} \leq 1} W_p(f_\#\mu, f_\#\nu),$$

where $f_\#\mu$ and $f_\#\nu$ are the pushforward measures of $\mu$ and $\nu$ by $f : \mathcal{B} \to \mathbb{R}$, respectively.

In particular, for dot product kernel $K(x, y) = x^{\mathrm{T}}y$, the RKHS $\mathcal{H} = \{f : f(x) = x^{\mathrm{T}}\beta, \exists \beta \in \mathbb{R}^d\}$. In this case, the KMS Wasserstein distance reduces to the max-sliced Wasserstein distance [16]. A more flexible choice is the Gaussian kernel $K(x, y) = \exp(-\frac{1}{2\sigma^2}\|x - y\|_2^2)$, where $\sigma > 0$ denotes the temperature hyper-parameter. In this case, the function class $\mathcal{H}$ satisfies the *universal property* as it is dense in the continuous function class with respect to the $\infty$-type functional norm. It is easy to see the following theorem holds.

**Theorem 2.4** (Metric Property of KMS). $\mathcal{KMS}_p(\cdot, \cdot)$ *satisfies the triangle inequality. If additionally assuming $\mathcal{H}$ is a universal RKHS, $\mathcal{KMS}_p(\mu, \nu) = 0$ if and only if $\mu = \nu$.*

**Example 2.5.** We present a toy example that highlights the flexibility of the KMS Wasserstein distance. Fig.1(a) displays a scatter plot of the circle dataset, which consists of two groups of samples distributed along inner and outer circles, perturbed by Gaussian noise. Since the data exhibit a nonlinear structure, distinguishing these groups using a linear projection is challenging. As shown in Fig.1(b), the density plot of the projected samples using the MS Wasserstein distance with a linear projector is not sufficiently discriminative. In contrast, Fig. 1(c) demonstrates that the KMS Wasserstein distance is better suited for distinguishing these two groups. Intuitively, the optimal nonlinear projector should take the form $f^*(x) \propto \|x\|_2^c$ for some scalar $c > 0$, as illustrated in Fig.1(d). We plot the nonlinear projector by computing the empirical KMS Wasserstein distance, shown in Fig.1(e), which closely resembles the circular landscape depicted in Fig. 1(d). This result demonstrates that the KMS Wasserstein distance provides a data-driven, non-parametric nonlinear projector capable of effectively distinguishing distinct data groups.

Given the RKHS $\mathcal{H}$, let the *canonical feature map* that embeds data to $\mathcal{H}$ as

$$\Phi : \mathcal{B} \to \mathcal{H}, \qquad x \mapsto \Phi(x) = K_x. \tag{1}$$

Define the functional $u_f : \mathcal{H} \to \mathbb{R}$ by $u_f(g) = \langle f, g \rangle_{\mathcal{H}}$ for any $g \in \mathcal{H}$, which can be viewed as a linear projector that maps data from the Hilbert space $\mathcal{H}$ to the real line. In

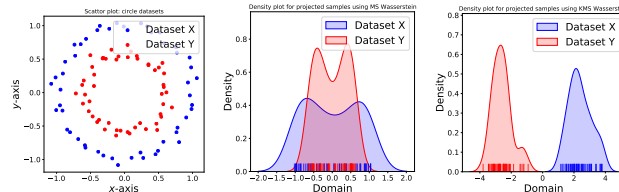

(a) Scatter plot of circle dataset  (b) Density plot using MS Wasserstein  (c) Density plot using KMS Wasserstein

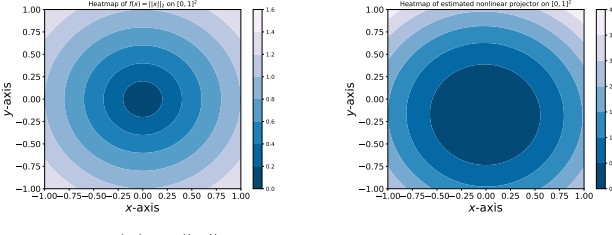

(d) Plot of $f(x) = \|x\|_2, x \in [0,1]^2$  (e) Plot of estimated projector using KMS Wasserstein

*Figure 1.* Results on a 2-dimensional toy example.

light of this, for two probability measures $\mu$ and $\nu$ on $\mathcal{H}$, we define the MS $p$-Wasserstein distance

$$\mathcal{MS}_p(\mu, \nu) = \sup_{f \in \mathcal{H}: \, \|f\|_{\mathcal{H}} \leq 1} W_p\big((u_f)_\#\mu, (u_f)_\#\nu\big), \tag{2}$$

where $(u_f)_\#\mu$ denotes the pushforward measure of $\mu$ by the map $u_f$, i.e., if $\mu$ is the distribution of a random element $X$ of $\mathcal{H}$, then $(u_f)_\#\mu$ is the distribution of the random variable $u_f(X) = \langle f, X \rangle_{\mathcal{H}}$, and $(u_f)_\#\nu$ is defined likewise. In the following, we show that the KMS Wasserstein distance in Definition 2.3 can be reformulated as the MS Wasserstein distance between two distributions on (infinite-dimensional) Hilbert space.

**Remark 2.6** (Reformulation of KMS Wasserstein). By the reproducing property, we can see that $f(x) = \langle f, K_x \rangle_{\mathcal{H}} = u_f(\Phi(x))$, which implies $f = u_f \circ \Phi$. As a consequence,

$$
\begin{aligned}
&\mathcal{KMS}_p(\mu, \nu) \\
&= \sup_{f \in \mathcal{H}: \, \|f\|_{\mathcal{H}} \leq 1} W_p\big((u_f)_\#\big(\Phi_\#\mu\big), (u_f)_\#\big(\Phi_\#\nu\big)\big) \\
&= \mathcal{MS}_p\big(\Phi_\#\mu, \Phi_\#\nu\big).
\end{aligned}
\tag{3}
$$

In other words, the KMS Wasserstein distance first maps data points into the infinite-dimensional Hilbert space $\mathcal{H}$ through the canonical feature map $\Phi$, and next finds the linear projector to maximally distinguish data from two populations. Compared with the traditional MS Wasserstein distance [16] that performs linear projection in $\mathbb{R}^d$, KMS Wasserstein distance is a more flexible notion.

**Remark 2.7** (Connections with Kernel PCA). Given data points $x_1, \ldots, x_n$ on $\mathcal{B}$, denote by $\widehat{\mu}_n$ the corresponding empirical distribution. Assume $\frac{1}{n} \sum_{i \in [n]} \Phi(x_i) = 0$, since

otherwise one can center those data points as a preprocessing step. Kernel PCA [52] is a popular tool for nonlinear dimensionality reduction. When seeking the first principal nonlinear projection function $f$, Mairal and Vert [51] presents the following reformulation of kernel PCA:

$$\underset{f\in\mathcal{H}:\ \|f\|_{\mathcal{H}}\leq 1}{\arg\max}\ \mathrm{Var}\Big((u_f)_{\#}(\Phi_{\#}\widehat{\mu}_n)\Big), \qquad (4)$$

where $\mathrm{Var}(\cdot)$ denotes the variance of a given probability measure. In comparison, the KMS Wasserstein distance aims to find the optimal nonlinear projection that distinguishes two populations and replaces the variance objective in (4) with the Wasserstein distance between two projected distributions in (3). Also, kernel PCA is a special case of KMS Wasserstein by taking $p=2$, $\mu\equiv\widehat{\mu}_n,\nu\equiv\delta_0$ in (3).

**Notations.** Let $\langle\cdot,\cdot\rangle$ denote the inner product operator. For any positive integer $n$, denote $[n]=\{1,2,\ldots,n\}$. Define $\Gamma_n$ as the set

$$\left\{\pi\in\mathbb{R}_+^{n\times n}:\ \sum_{i=1}^n\pi_{i,j}=\frac{1}{n},\sum_{j=1}^n\pi_{i,j}=\frac{1}{n},\forall i,j\in[n]\right\}.$$
$$(5)$$

Let $\mathrm{Conv}(P)$ denote a convex hull of the set $P$, and $\mathbb{S}_n^+$ denote the set of positive semidefinite matrices of size $n\times n$. We use $\widetilde{\mathcal{O}}(\cdot)$ as a variant of $\mathcal{O}(\cdot)$ to hide logarithmic factors.

## 3. Statistical Guarantees

Suppose samples $x^n := \{x_i\}_{i\in[n]}$ and $y^n := \{y_i\}_{i\in[n]}$ are given and follow distributions $\mu,\nu$, respectively. Denote by $\widehat{\mu}_n$ and $\widehat{\nu}_n$ the corresponding empirical distributions from samples $x^n$ and $y^n$. In this section, we provide a finite-sample guarantee on the $p$-KMS Wasserstein distance between $\widehat{\mu}_n$ and $\widehat{\nu}_n$ with $p\in[1,\infty)$. This guarantee can be helpful for KMS Wasserstein distance-based hypothesis testing that has been studied in [75]: Suppose one aims to build a non-parametric test to distinguish two hypotheses $H_0:\ \mu=\nu$ and $H_1:\ \mu\neq\nu$. Thus, it is crucial to control the high-probability upper bound of $\mathcal{KMS}_p(\widehat{\mu}_n,\widehat{\nu}_n)$ under $H_0$ as it serves as the critical value to determine whether $H_0$ is rejected or not. We first make the following assumption on the kernel.

**Assumption 3.1.** There exists some constant $A>0$ such that the kernel $K(\cdot,\cdot)$ satisfies $\sqrt{K(x,x)}\leq A,\forall x\in\mathcal{B}$.

Assumption 3.1 is standard in the literature (see, e.g., [28]), and is quite mild: Gaussian kernel $K(x,y)=\exp(-\|x-y\|_2^2/\sigma^2)$ naturally fits into this assumption. For dot product kernel $K(x,y)=x^{\mathrm{T}}y$, if we assume the support $\mathcal{B}$ has a finite diameter, this assumption can also be satisfied. Define the critical value

$$\Delta(n,\alpha)=4A\Big(C+4\sqrt{\log\frac{2}{\alpha}}\Big)^{1/p}\cdot n^{-1/(2p)},$$

where $C\geq 1$ is a universal constant. We show the following finite-sample guarantees on the KMS $p$-Wasserstein distance.

**Theorem 3.2** (Finite-Sample Guarantees). *Fix $p\in[1,\infty)$, level $\alpha\in(0,1)$, and suppose Assumption 3.1 holds.*

(I) *(One-Sample Guarantee) With probability at least $1-\alpha$, it holds that $\mathcal{KMS}_p(\widehat{\mu}_n,\mu)\leq\frac{1}{2}\Delta(n,\alpha)$.*

(II) *(Two-Sample Guarantee) With probability at least $1-\alpha$, it holds that*

$$\mathcal{KMS}_p(\widehat{\mu}_n,\widehat{\nu}_n)\leq\mathcal{KMS}_p(\mu,\nu)+\Delta(n,\alpha).$$

The proof of Theorem 3.2 is provided in Appendix B. The dimension-free upper bound $\Delta(n,\alpha)=O(n^{-1/(2p)})$ is optimal in the worst case. Indeed, in the one-dimension case $\mathcal{B}=[0,1]$ and $K(x,y)=xy$, the kernel max-sliced Wasserstein distance $\mathcal{KMS}_p$ coincides with the classical Wasserstein distance $W_p$. In this case, it is easy to see that if $\mu=(\delta_0+\delta_1)/2$ is supported on the two points 0 and 1, the expectation of $\mathcal{KMS}(\widehat{\mu}_n,\widehat{\nu}_n)$ is of order $n^{-1/(2p)}$ [22]. We also compare this bound with other OT divergences in Appendix A.

We design a two-sample test $\mathcal{T}_{\mathrm{KMS}}$ such that $H_0$ is rejected if $\mathcal{KMS}_p(\widehat{\mu}_n,\widehat{\nu}_n)>\Delta(n,\alpha)$. By Theorem 3.2, we have the following performance guarantees of $\mathcal{T}_{\mathrm{KMS}}$.

**Corollary 3.3** (Testing Power of $\mathcal{T}_{\mathrm{KMS}}$). *Fix a level $\alpha\in(0,1/2)$, $p\in[1,\infty)$, and suppose Assumption 3.1 holds. Then the following result holds:*

(I) *(Risk): The type-I risk of $\mathcal{T}_{\mathrm{KMS}}$ is at most $\alpha$;*

(II) *(Power): Under $H_1:\ \mu\neq\nu$, suppose the sample size $n$ is sufficiently large such that $\varrho_n := \mathcal{KMS}_p(\mu,\nu)-\Delta(n,\alpha)>0$, the power of $\mathcal{T}_{\mathrm{KMS}}$ is at least $1-c\cdot n^{-1/(2p)}$, where $c$ is a constant depending on $A,C,p,\varrho_n$.*

The proof of Corollary 3.3 is provided in Appendix B. It is also noteworthy that the performance of $\mathcal{T}_{\mathrm{KMS}}$ still has a non-negligible dependence on the data dimension. Let us follow Ramdas et al. [64] to consider the fair alternative that the discrepancy between $\mu$ and $\nu$ remains constant as the dimension increases. Then, the term $\varrho_n$ decreases as the dimension increases. To ensure that the assumption $\varrho_n>0$ holds in Corollary 3.3, the sample size $n$ should increase to maintain satisfactory testing power when the data dimension increases.

**Remark 3.4** (Comparison with Maximum Mean Discrepancy (MMD)). MMD has been a popular kernel-based tool to quantify the discrepancy between two probability measures (see, e.g., [6, 23, 28, 37, 48, 53, 66, 67, 69, 72]), which,

for any two probability distributions $\mu$ and $\nu$, is defined as

$$\text{MMD}(\mu, \nu) = \max_{\substack{f \in \mathcal{H}, \\ \|f\|_{\mathcal{H}} \leq 1}} \mathbb{E}_\mu[f] - \mathbb{E}_\nu[f]$$

$$= \max_{\substack{f \in \mathcal{H}, \\ \|f\|_{\mathcal{H}} \leq 1}} \overline{(u_f)_\# (\Phi_\# \mu)} - \overline{(u_f)_\# (\Phi_\# \nu)}, \quad (6)$$

where $\overline{\xi}$ denotes the mean of a given probability measure $\xi$. The empirical (biased) MMD estimator also exhibits dimension-free finite-sample guarantee as in Theorem 3.2: it decays in the order of $\mathcal{O}(n^{-1/2})$, where $n$ is the number of samples. However, the KMS Wasserstein distance is more flexible as it replaces the mean difference objective in (6) by the Wasserstein distance, which naturally incorporates the geometry of the sample space and is suitable for hedging against adversarial data perturbations [24].

## 4. Computing $2$-KMS Wasserstein Distance

Let $\widehat{\mu}_n$ and $\widehat{\nu}_n$ be two empirical distributions supported on $n$ points, i.e., $\widehat{\mu}_n = \frac{1}{n} \sum_i \delta_{x_i}$, $\widehat{\nu}_n = \frac{1}{n} \sum_j \delta_{y_j}$, where $\{x_i\}_i, \{y_j\}_j$ are data points in $\mathbb{R}^d$. This section focuses on the computation of 2-KMS Wasserstein distance between these two distributions. By Definition 2.3 and monotonicity of square root function, it holds that

$$\mathcal{KMS}_2(\widehat{\mu}_n, \widehat{\nu}_n)$$

$$= \left( \max_{f \in \mathcal{H}, \, \|f\|_{\mathcal{H}}^2 \leq 1} \left\{ \min_{\pi \in \Gamma_n} \sum_{i,j \in [n]} \pi_{i,j} |f(x_i) - f(y_j)|^2 \right\} \right)^{1/2},$$

$$\text{(KMS)}$$

where $\Gamma_n$ is defined in (5).

Although the outer maximization problem is a *functional optimization* that contains uncountably many parameters, one can apply the representer theorem (see below) to reformulate Problem (KMS) as a finite-dimensional optimization.

**Theorem 4.1** (Theorem 1 in [75]). *There exists an optimal solution to* (KMS), *denoted as* $\widehat{f}$, *such that for any* $z$,

$$\widehat{f}(z) = \sum_{i=1}^n a_{x,i} K(z, x_i) - \sum_{i=1}^n a_{y,i} K(z, y_i), \quad (7)$$

*where* $a_x = (a_{x,i})_{i \in [n]}, a_y = (a_{y,i})_{i \in [n]}$ *are coefficients to be determined.*

Define gram matrix $K(x^n, x^n) = (K(x_i, x_j))_{i,j \in [n]}$ and other gram matrices $K(x^n, y^n)$, $K(y^n, x^n)$, $K(y^n, y^n)$ likewise, then define the concatenation of gram matrics

$$G = \begin{pmatrix} K(x^n, x^n) & -K(x^n, y^n) \\ -K(y^n, x^n) & K(y^n, y^n) \end{pmatrix} \in \mathbb{R}^{2n \times 2n}. \quad (8)$$

Assume $G$ is positive definite [1] such that it admits the

---

[1] In Appendix C, we provide a sufficient condition that ensures $G$ is positive definite.

Cholesky decomposition $G^{-1} = UU^\mathrm{T}$. Define

$$M'_{i,j} = \begin{pmatrix} (K(x_i, x_l) - K(y_j, x_l))_{l \in [n]} \\ (K(y_j, y_l) - K(x_i, y_l))_{l \in [n]} \end{pmatrix} \in \mathbb{R}^{2n}.$$

and the vector $M_{i,j} = U^\mathrm{T} M'_{i,j}$. By substituting the expression (7) into (KMS) and calculation (see Appendix D), we obtain the exact reformulation of (KMS):

$$\max_{\omega \in \mathbb{R}^{2n}: \, \|\omega\|_2 = 1} \left\{ \min_{\pi \in \Gamma_n} \sum_{i,j} \pi_{i,j} (M_{i,j}^\mathrm{T} \omega)^2 \right\}. \quad (9)$$

Here, we omit taking the square root of the optimal value of the max-min optimization problem for simplicity of presentation. Since Problem (9) is a non-convex program, it is natural to question its computational hardness. The following gives an *affirmative* answer, whose proof is provided in Appendix E.

**Theorem 4.2** (NP-hardness). *Problem* (9) *is NP-hard for the worst-case instances of* $\{M_{i,j}\}_{i,j}$.

The proof idea of Theorem 4.2 is to find an instance of $\{M_{i,j}\}_{i,j}$ that depends on a generic collection of $n$ vectors $\{A_i\}_i$ such that solving (9) is at least as difficult as solving the fair-PCA problem [65] with rank-1 matrices (or fair beamforming problem [68]) and has been proved to be NP-hard [68]. Interestingly, the computational hardness of the MS Wasserstein distance arises from both the data dimension $d$ and the sample size $n$, whereas that of the KMS Wasserstein distance arises from the sample size $n$ only.

To tackle the computational challenge of solving (9), in the subsequent subsections, we present an SDR formula and propose an efficient first-order algorithm to solve it. Next, we analyze the computational complexity of our proposed algorithm and the theoretical guarantees on SDR.

### 4.1. Semidefinite Relaxation with Efficient Algorithms

We observe the simple reformulation of the objective in (9):

$$\sum_{i,j} \pi_{i,j} (M_{i,j}^\mathrm{T} \omega)^2 = \sum_{i,j} \pi_{i,j} \langle M_{i,j} M_{i,j}^\mathrm{T}, \omega \omega^\mathrm{T} \rangle.$$

Inspired by this relation, we use the change of variable approach to optimize the rank-1 matrix $S = \omega \omega^\mathrm{T}$, i.e., it suffices to consider the equivalent reformulation of (9):

$$\max \left\{ F(S) : \, S \in \mathbb{S}_+^{2n}, \text{Trace}(S) = 1, \text{rank}(S) = 1 \right\}$$

where $F(S) = \min_{\pi \in \Gamma_n} \sum_{i,j} \pi_{i,j} \langle M_{i,j} M_{i,j}^\mathrm{T}, S \rangle.$

$$(10)$$

An efficient SDR is to drop the rank-1 constraint to consider the semidefinite program (SDP):

$$\max_{S \in \mathcal{S}_{2n}} F(S),$$

$$\text{where} \quad \mathcal{S}_{2n} = \left\{ S \in \mathbb{S}_+^{2n} : \, \text{Trace}(S) = 1 \right\}. \quad \text{(SDR)}$$

**Remark 4.3** (Connection with [80])**.** We highlight that Xie and Xie [80] considered the same SDR heuristic to compute the MS 1-Wasserstein distance. However, the authors therein apply the interior point method to solve a large-scale SDP, which has expansive complexity $\mathcal{O}(n^{6.5}\text{polylog}(\frac{1}{\delta}))$ (up to $\delta$-accuracy) [5]. In the following, we present a first-order method that exhibits much smaller complexity $\widetilde{\mathcal{O}}\left(n^2\delta^{-3}\right)$ in terms of the problem size $n$ (see Theorem 4.4). Besides, theoretical guarantees on the solution from SDR have not been explored in [80], and we are the first literature to provide these results.

The constraint set $\mathcal{S}_{2n}$ is called the *Spectrahedron* and admits closed-form Bregman projection. Inspired by this, we propose an inexact mirror ascent algorithm to solve (SDR). Its high-level idea is to iteratively construct an inexact gradient estimator and next perform the mirror ascent on iteration points. By properly balancing the trade-off between the bias and cost of querying gradient oracles, this type of algorithm is guaranteed to find a near-optimal solution [29–32].

We first discuss how to construct supgradient estimators of $F$. By Danskin's theorem [7],

$$\partial F(S) = \text{Conv}\left\{\sum_{i,j}\pi_{i,j}^*(S)M_{i,j}M_{i,j}^{\mathrm{T}} : \pi^*(S) \in \Gamma_n(S)\right\},$$

where $\Gamma_n(S)$ denotes the set of optimal solutions to the following OT problem:

$$\min_{\pi \in \Gamma_n} \sum_{i,j}\pi_{i,j}\langle M_{i,j}M_{i,j}^{\mathrm{T}}, S\rangle. \tag{11}$$

The main challenge of constructing a supgradient estimator is to compute an optimal solution $\pi^*(S) \in \Gamma_n(S)$. Since computing an exactly optimal solution is too expensive, we derive its near-optimal estimator, denoted as $\widehat{\pi}$, and practically use the following supgradient estimator:

$$v(S) = \sum_{i,j}\widehat{\pi}_{i,j}M_{i,j}M_{i,j}^{\mathrm{T}}. \tag{12}$$

We adopt the stochastic gradient-based algorithm with Katyusha momentum in [82] to compute a $\epsilon$-optimal solution $\widehat{\pi}$ to (11). It achieves the state-of-the-art complexity $\widetilde{\mathcal{O}}\left(n^2\epsilon^{-1}\right)$. See the detailed algorithm in Appendix F. Next, we describe the main algorithm for solving (SDR). Define the (negative) von Neumann entropy $h(S) = \sum_{i\in[2n]}\lambda_i(S)\log\lambda_i(S)$, where $\{\lambda_i(S)\}_i$ are the eigenvalues of $S$, and define the von Neumann Bregman divergence

$$V(S_1, S_2) = h(S_1) - h(S_2) - \langle S_1 - S_2, \nabla h(S_2)^{\mathrm{T}}\rangle$$
$$= \text{Trace}(S_1\log S_1 - S_1\log S_2).$$

Iteratively, we update $S_{k+1}$ by performing mirror ascent with constant stepsize $\gamma > 0$:

$$S_{k+1} = \arg\max_{S\in\mathcal{S}_{2n}} \gamma\langle v(S_k), S\rangle + V(S, S_k),$$

---

**Algorithm 1** Inexact Mirror Ascent for solving (SDR)

1: **Input:** Max iterations $T$, initial guess $S_1$, tolerance $\epsilon$, constant stepsize $\gamma$.
2: **for** $k = 1, \ldots, T-1$ **do**
3:     Obtain a $\epsilon$-optimal solution (denoted as $\widehat{\pi}$) to (11)
4:     Construct inexact supgradient $v(S_k)$ by (12)
5:     Perform mirror ascent by (13)
6: **end for**
7: **Return** $\widehat{S}_{1:T} = \frac{1}{T}\sum_{k=1}^{T}S_k$

---

which admits the following closed-form update:

$$\widetilde{S}_{k+1} = \exp\left(\log S_k + \gamma v(S_k)\right),$$
$$S_{k+1} = \frac{\widetilde{S}_{k+1}}{\text{Trace}(\widetilde{S}_{k+1})}. \tag{13}$$

The $\exp(\cdot)$ and $\log(\cdot)$ operators above refer to matrix exponential and matrix logarithm, respectively. As $S_k$ is always positive semidefinite, the update in (13) incurs the computational cost $\mathcal{O}(n^3)$ in the worst case. The general procedure for solving (SDR) is summarized in Algorithm 1.

### 4.2. Theoretical Analysis

In this subsection, we establish the complexity and performance guarantees for solving (SDR). Since the constraint set $\mathcal{S}_{2n}$ is compact and the objective in (SDR) is continuous, an optimal solution, denoted by $S^*$, is guaranteed to exist with a finite optimal value. A feasible solution $\widehat{S} \in \mathcal{S}_{2n}$ is said to be $\delta$-optimal if it satisfies the condition $F(\widehat{S}) - F(S^*) \leq \delta$. Define the constant $C = \max_{i,j}\|M_{i,j}\|_2^2$.

**Theorem 4.4** (Complexity Bound)**.** *Fix the precision $\delta > 0$ and specify hyper-parameters*

$$T = \left\lceil\frac{16C^2\log(2n)}{\delta^2}\right\rceil, \qquad \epsilon = \frac{\delta}{4}, \qquad \gamma = \frac{\log(2n)}{C\sqrt{T}}.$$

*Then, the complexity of Algorithm 1 for finding $\delta$-optimal solution to (SDR) is*

$$\widetilde{\mathcal{O}}\left(n^2\delta^{-2}\cdot\max(n, \delta^{-1})\right).$$

Next, we analyze the quality of the solution to (SDR). Recall the exact reformulation (9) requires that the optimal solution to be rank-1 while the tractable relaxation (SDR) does not enforce such a constraint. Therefore, it is of interest to provide theoretical guarantees on the low-rank solution of (SDR), i.e., we aim to find the smallest integer $k \geq 1$ such that there exists an optimal solution to (SDR) that is at most rank-$k$. The integer $k$ is called a rank bound on (SDR), which is characterized in the following theorem.

**Theorem 4.5** (Rank Bound on (SDR))**.** *There exists an optimal solution to (SDR) of rank at most $k \triangleq 1 +$*

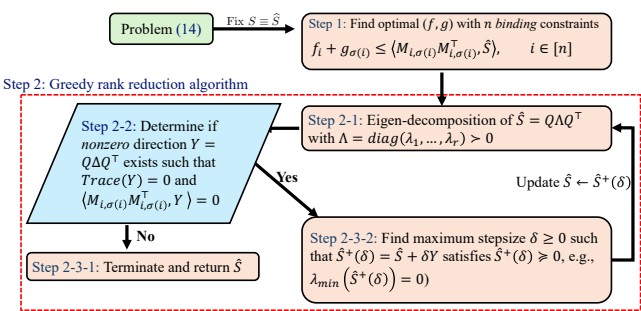

*Figure 2.* Diagram of the rank reduction algorithm. Here $\sigma(\cdot)$ denotes the permutation operator on $[n]$, Step 1 can be implemented using the Hungarian algorithm [41], and Step 2-2 finds a direction that lies in the null space of the constraint of Problem (14).

$$\left\lfloor \sqrt{2n + \tfrac{9}{4}} - \tfrac{3}{2} \right\rfloor. \text{ As a result,}$$

$$Optval(9) = \max_{S \in \mathbb{S}_{+}^{2n}, Trace(S)=1, rank(S)=1} F(S)$$

$$\leq Optval(SDR) \leq \max_{S \in \mathbb{S}_{+}^{2n}, Trace(S)=1, rank(S)=k} F(S).$$

The trivial rank bound on (SDR) should be $2n$, as the matrix $S$ is of size $2n \times 2n$. Theorem 4.5 provides a novel rank bound that is significantly smaller than the trivial one.

*Proof Sketch of Theorem 4.5.* We first reformulate (SDR) by taking the dual of the inner OT problem:

$$\max_{\substack{S \in \mathcal{S}_{2n} \\ f,g \in \mathbb{R}^n}} \left\{ \frac{1}{n} \sum_{i=1}^{n} (f_i + g_i) : f_i + g_j \leq \langle M_{i,j} M_{i,j}^{\mathrm{T}}, S \rangle, \forall i, j \right\}.$$

$$(14)$$

By Birkhoff's theorem [8] and complementary slackness of OT, there exists an optimal solution of $(f, g)$ such that at most $n$ constraints of (14) are binding, and with such an optimal choice, one can adopt the convex geometry analysis from [43, 44] to derive the desired rank bound for any feasible extreme point of variable $S$. As the set of optimal solutions of (SDR) has a non-empty intersection with the set of feasible extreme points, the desired result holds. □

It is noteworthy that Algorithm 1 only finds a near-optimal solution $\widehat{S}_{1:T}$ of (SDR), which is not guaranteed to satisfy the rank bound in Theorem 4.5. To fill the gap, we develop a rank-reduction algorithm that further converts $\widehat{S}_{1:T}$ to the feasible solution that maintains the desired rank bound. [2]

---

[2]Our motivation for rank-reduction is that it influences the quality of approximation to the original non-convex optimization problem (KMS). Recall that globally solving (KMS) involves a rank-1 constraint, and (SDR) relaxes it. When the solution to (SDR) is of rank > 1, we construct a feasible rank-1 solution by taking the leading eigenvector. However, if the solution to (SDR) is already low-rank, the rounding procedure typically yields a feasible solution that is closer to the ground-truth rank-1 optimum.

See the general diagram that outlines this algorithm in Fig. 2 and the detailed description in Appendix J. We also provide its complexity analysis in the following theorem, though in numerical study the complexity is considerably smaller than the theoretical one.

**Theorem 4.6.** *The rank reduction algorithm in Fig. 2 satisfies that* (I) *for a $\delta$-optimal solution to (SDR), it outputs another $\delta$-optimal solution with rank at most $k$;* (II) *its worst-case complexity is $\mathcal{O}(n^5)$.*

Additionally, by adopting the proof from Luo et al. [50], we show the optimality gap guarantee in Theorem 4.7. Although the approximation ratio seems overly conservative, we find (SDR) has good numerical performance.

**Theorem 4.7** (Relaxation Gap of (SDR)). *Denote by $\varepsilon = 4 \cdot (0.33)^3$ an universal constant. Then*

$$\varepsilon n^{-4} \cdot \mathrm{Optval}(\mathrm{SDR}) \leq \mathrm{Optval}(\mathrm{KMS}) \leq \mathrm{Optval}(\mathrm{SDR}).$$

## 5. Numerical Study

This section presents experiment results for KMS 2-Wasserstein distance that is solved using SDR with first-order algorithm and rank reduction (denoted as `SDR-Efficient`). Baseline approaches include the block coordinate descent (`BCD`) algorithm [75], which finds stationary points of KMS 2-Wasserstein, and using interior point method (IPM) by off-the-shelf solver `cvxpy` [19] for solving SDR relaxation (denoted as `SDR-IPM`). Each instance is allocated a maximum time budget of one hour. All experiments were conducted on a MacBook Pro with an Intel Core i9 2.4GHz and 16GB memory. Unless otherwise stated, error bars are reproduced using 20 independent trials. Throughout the experiments, we specify the kernel as Gaussian, with the bandwidth being the median of pairwise distances between data points. Other details and extra numerical studies can be found in Appendices K and L.

**Computational Time and Solution Quality.** We first compare our approach to baseline methods in terms of running time and solution quality. The quality of a given nonlinear projector is assessed by projecting testing data points from two groups and calculating their 2-Wasserstein distance. The experimental results, shown in the top of Fig. 3, indicate that for small sample size instances, `SDR-IPM` requires significantly more time than the other two approaches. Additionally, for small sample size instances, the running time of `BCD` is slightly shorter than that of `SDR-Efficient`. However, for larger instances, `BCD` outperforms `SDR-Efficient` in terms of running time. This observation aligns with our theoretical analysis, which shows that the complexity of SDR is $\tilde{\mathcal{O}}(n^2 \delta^{-2} \max(n, \delta^{-1}))$, lower than the complexity of `BCD` [75], which is $\tilde{\mathcal{O}}(n^3 \delta^{-3})$. The plots in the bottom of

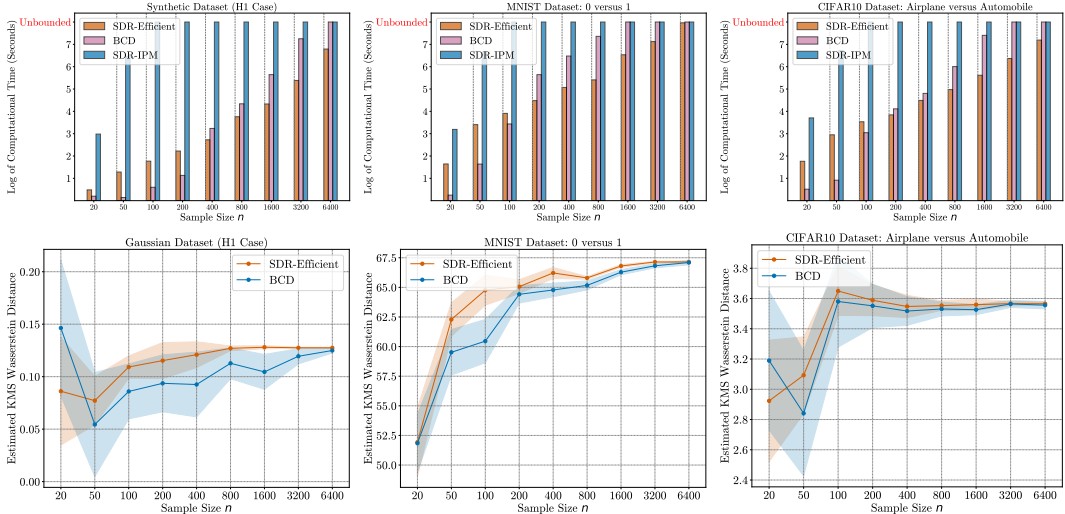

*Figure 3.* Comparison of `SDR-Efficient` with the baseline methods `SDR-IPM` and `BCD` in terms of computational time and solution quality. The columns, from left to right, correspond to the synthetic Gaussian dataset (100-dimensional), `MNIST`, and `CIFAR-10`. The top plots display the computational time, where the $y$-axis is labeled as "unbounded" if the running time exceeds the 1-hour time limit. The bottom plots present the estimated KMS 2-Wasserstein distance for each method.

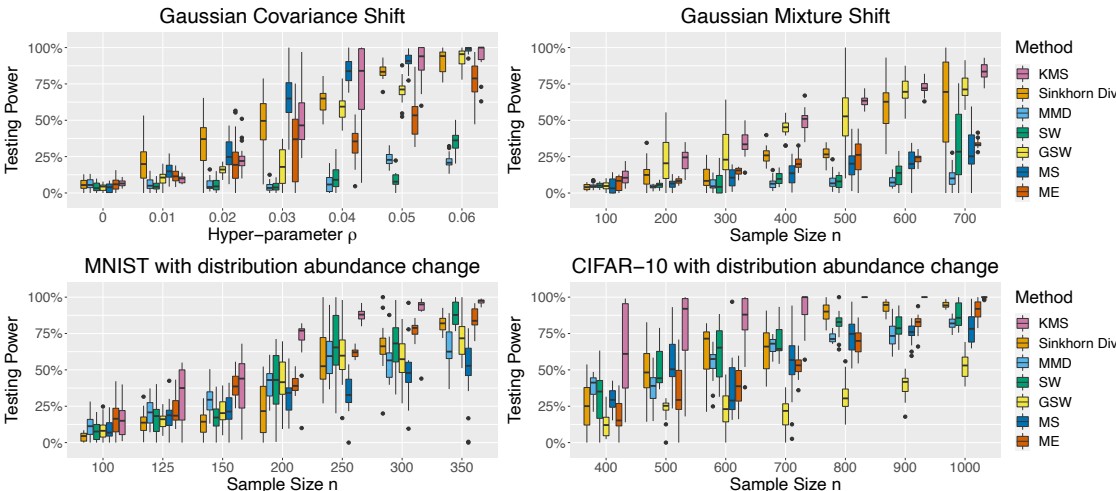

*Figure 4.* Testing power with a controlled type-I error rate of 0.05 across four datasets. Figures from left to right correspond to (a) Gaussian covariance shift, (b) Gaussian mixture distribution shift, (c) MNIST, and (d) CIFAR-10 with distribution abundance changes.

Fig. 3 show the quality of methods `SDR-Efficient` and `BCD`. We find the performance of solving SDR outperforms `BCD`, as indicated by its larger means and smaller variations. One possible explanation is that `BCD` is designed to find a local optimum solution for the original non-convex problem, making it highly sensitive to the initial guess and potentially less effective in achieving optimal performance.

**High-dimensional Hypothesis Testing.** Then we validate the performance of KMS 2-Wasserstein distance for high-dimensional two-sample testing using both synthetic and real datasets. Baseline approaches include the two-

sample testing with other statistical divergences, such as (i) Sinkhorn divergence (`Sinkhorn Div`) [27], (ii) maximum mean discrepancy with Gaussian kernel and median bandwidth heuristic (`MMD`) [28], (iii) sliced Wasserstein distance (`SW`) [10], (iv) generalized-sliced Wasserstein distance (`GSW`) [38], (v) max-sliced Wasserstein distance (`MS`) [16], and (vi) optimized mean-embedding test (`ME`) [36]. The baselines `KMS, MS, ME` partition data into training and testing sets, learn parameters from the training set, and evaluate the testing power on the testing set. Other baselines utilize both sets for evaluation. Synthetic datasets include the high-dimensional Gaussian distributions

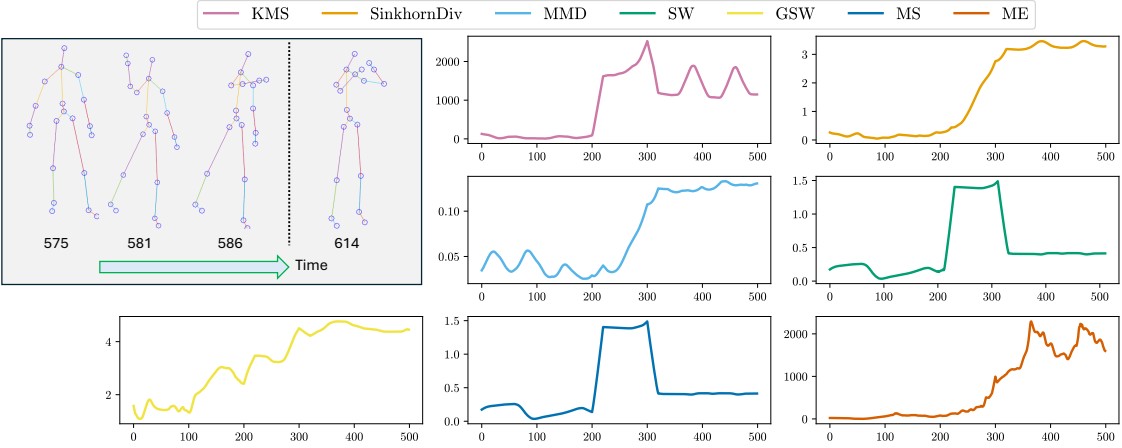

Figure 5. Top Left: Illustration of sequential data before and after the change-point. Remaining: Testing statistics computed from our and baseline approaches.

with covariance shift, or Gaussian mixture distributions. Real datasets include the MNIST [15] and CIFAR-10 [40] with changes in distribution abundance. The type-I error is controlled within 0.05 for all methods.

We report the testing power for all these approaches and datasets in Fig. 4. For the Gaussian covariance shift scenario the MS method achieves the best performance, which can be explained by the fact that linear mapping is optimal to separate the high-dimensional Gaussian distributions. For the other three scenarios, our approach has the superior performance compared with those baselines.

Table 1. Detection delay of various methods with controlled false alarm rate $\alpha = 0.05$. Mean and standard deviation (std) are calculated based on data from 10 different users.

| Method | KMS | Sinkhorn Div | MMD | SW | GSW | MS | ME |
|--------|-----|--------------|-----|-----|-----|-----|-----|
| Mean | **11.4** | 16.5 | 50.6 | 17.2 | 12.9 | 17.8 | 65.4 |
| Std | 5.56 | **4.4** | 39.5 | 8.7 | 6.4 | 9.2 | 25.7 |

**Human Activity Detection.** We evaluate the performance of the KMS Wasserstein distance in detecting human activity transitions as quickly as possible using MSRC-12 Kinect gesture dataset [21]. After preprocessing, the dataset consists of 10 users, each with 80 attributes, performing the action throwing/lifting before/after the change-point at time index 600. We employ a sliding window approach [81] with a false alarm rate of 0.01 to construct the test statistic at each time index, which increases significantly when a change-point is detected. Figure 5 illustrates the test statistics generated by our method compared to baseline approaches. The experimental results, summarized in Table 1, show that our method achieves superior performance.

**Generative Modeling.** Finally, we examine the perfor-

mance of various statistical divergences in generative modeling by following the experiment setup in [39]. We compute the Fréchet Inception Distance (FID) scores of generated fake images using a convolutional neural network trained on real images to achieve satisfactory classification accuracy. Lower FID score suggests better generative modeling performance. Experiment results are reported in Figure 6, from which we can see that KMS Wasserstein distance can learn high-dimensional distributions using data.

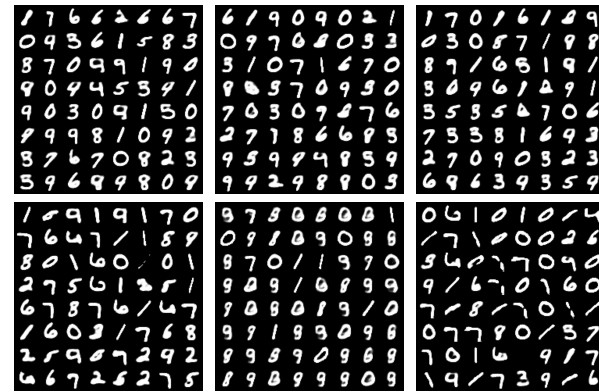

Figure 6. Plots from left to right correspond to fake images obtained by generative models using (a) KMS Wasserstein, (b) Sinkhorn divergence, (c) MMD, (d) Sliced Wasserstein, (e) Generalized Sliced Wasserstein, and (f) Max-Sliced Wasserstein. The FID scores are 105.98, 114.36, 113.08, 113.92, 128.07, and 115.21, respectively.

# 6. Concluding Remarks

In this paper, we provided statistical and computational guarantees of the KMS Wasserstein distance. Our numerical study validated our theoretical results and highlighted the exceptional performance of the KMS Wasserstein distance.

## Acknowledgments

This work is partially supported by DMS-2134037.

## Impact Statement

This is a theoretical work on statistical and computational guarantees on the KMS Wasserstein distance. One of its societal impacts is its application to non-parametric two-sample testing. In practice, researchers can use two-sample testing to evaluate the effectiveness of medical treatments, discover economic disparities, detect anomaly observations, and more. We do not foresee any negative societal impact of this work.

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

## A. Comparison with Optimal Transport Divergences

*Table 2.* Time and sample complexity of empirical OT estimators in terms of the number of samples $n$. Here $\widehat{\mu}_n, \widehat{\nu}_n$ represent two empirical distributions based on i.i.d. $n$ sample points in $\mathbb{R}^d$. $p$ denotes the order of the metric defined in the cost function of standard Wasserstein distance.

| Reference | Estimator | Name | Time Complexity | Sample Complexity |
|---|---|---|---|---|
| [22] | $W_p(\widehat{\mu}_n, \widehat{\nu}_n)$ | Wasserstein distance | $\mathcal{O}(n^3 \log n)$ | $\mathcal{O}(n^{-1/d})$ |
| [26] | $\mathcal{S}_{p,\epsilon}(\widehat{\mu}_n, \widehat{\nu}_n)$ | Sinkhorn Divergence | $\mathcal{O}(n^2)$ per iteration | $\mathcal{O}(n^{-1/2}(1 + \epsilon^{d/2}))$ |
| [47] | $\mathcal{MS}_k(\widehat{\mu}_n, \widehat{\nu}_n)$ | Max Sliced Wasserstein distance with $k$-dimensional projector | $\tilde{\mathcal{O}}(n^2 d)$ | $\tilde{\mathcal{O}}(n^{-1/(\max\{k, 2p\})})$ |
| [57] | $\mathcal{SW}_p(\widehat{\mu}_n, \widehat{\nu}_n)$ | Sliced Wasserstein distance | $\tilde{\mathcal{O}}(nd)$ | $\tilde{\mathcal{O}}(n^{-1/(2p)})$ |
| [75] and this work | $\mathcal{KMS}_p(\widehat{\mu}_n, \widehat{\nu}_n)$ | Kernel Max Sliced Wasserstein Distance | $\tilde{\mathcal{O}}(n^2 d^3)$ | $\mathcal{O}(n^{-1/(2p)})$ |

In Table 2, we summarize the time and sample complexity of various OT divergences. Notably, the sample complexity of the standard Wasserstein distance suffers from the curse of dimensionality. While the Sinkhorn divergence mitigates this issue, its sample complexity depends on $\epsilon^{d/2}$, which can be prohibitively large when the regularization parameter $\epsilon$ is very small. For Max-Sliced (MS) and Sliced Wasserstein distances, their sample complexity is independent of the data dimension. However, they rely on linear projections to analyze data samples, which may not be optimal, particularly when the data exhibits a nonlinear low-dimensional structure. This limitation motivates the study of the KMS Wasserstein distance by Wang et al. [75] and in our work. In Example 2.5 and Section 5, we numerically validate this observation and demonstrate its practical advantages.

## B. Proof of Theorem 3.2 and Corollary 3.3

The proof in this part relies on the following technical results.

**Theorem B.1.** *(Finite-Sample Guarantee on MS 1-Wasserstein Distance on Hilbert Space, Adopted from [9, Corollary 2.8]) Let $\delta \in (0, 1]$, and $\mu$ be a probability measure on a separable Hilbert space $\mathcal{H}$ with $\int_{\mathcal{H}} \|x\| \, d\mu(x) < \infty$. Let $X_1, \ldots, X_n$ be i.i.d. random elements of $\mathcal{H}$ sampled according to $\mu$, and $\widehat{\mu}_n = \frac{1}{n} \sum_{i=1}^n \delta_{X_i}$, then it holds that*

$$\mathbb{E} \mathcal{MS}_1(\mu, \widehat{\mu}_n) \leq C \cdot \left( \int_{\mathcal{H}} \|x\|^{2+2\delta} \, d\mu(x) \right)^{1/(2+2\delta)} \cdot (\delta n)^{-1/2},$$

*where $C \geq 1$ is a universal constant.*

**Theorem B.2** (Functional Hoefffding Theorem [70, Theorem 3.26])**.** *Let $\mathcal{F}$ be a class of functions, each of the form $h : \mathcal{B} \to \mathbb{R}$, and $X_1, \ldots, X_n$ be samples i.i.d. drawn from $\mu$ on $\mathcal{B}$. For $i \in [n]$, assume there are real numbers $a_{i,h} \leq b_{i,h}$ such that $h(x) \in [a_{i,h}, b_{i,h}]$ for any $x \in \mathcal{B}, h \in \mathcal{F} \cup \{-\mathcal{F}\}$. Define*

$$L^2 = \sup_{h \in \mathcal{F} \cup \{-\mathcal{F}\}} \frac{1}{n} \sum_{i=1}^n (b_{i,h} - a_{i,h})^2.$$

*For all $\delta \geq 0$, it holds that*

$$\mathbb{P} \left\{ \sup_{h \in \mathcal{F}} \left| \frac{1}{n} \sum_{i=1}^n h(X_i) \right| \geq \mathbb{E} \left[ \sup_{h \in \mathcal{F}} \left| \frac{1}{n} \sum_{1}^n h(X_i) \right| \right] + \delta \right\} \leq \exp\left( -\frac{n\delta^2}{4L^2} \right).$$

We first show the one-sample guarantees for KMS $p$-Wasserstein distance.

**Proposition B.3.** *Fix $p \in [1, \infty)$, error probability $\alpha \in (0, 1)$, and suppose Assumption 3.1 holds. Let $C \geq 1$ be a universal constant. Then, we have the following results:*

(I) $\mathbb{E} \mathcal{KMS}_p(\mu, \widehat{\mu}_n) \leq A(2C^{1/p}) \cdot n^{-1/(2p)}$

(II) *With probability at least $1 - \alpha$, it holds that*

$$\mathcal{KMS}_p(\mu, \widehat{\mu}_n) \leq 2^{1-1/p} A \left( C + 4\sqrt{\log \frac{1}{\alpha}} \right)^{1/p} \cdot n^{-1/(2p)}.$$

*Proof of Proposition B.3.* Recall from (3) that

$$\mathcal{KMS}_p(\mu, \nu) = \mathcal{MS}_p\Big(\Phi_{\#}\mu, \Phi_{\#}\nu\Big).$$

Therefore, it suffices to derive one-sample guarantees for $\mathcal{MS}_p\Big(\Phi_{\#}\mu, \Phi_{\#}\widehat{\mu}_n\Big)$.

(I) Observe that under Assumption 3.1, we have

$$A^2 \geq K(x, x) = \langle K_x, K_x \rangle = \|K_x\|_{\mathcal{H}}^2,$$

and therefore $\|\Phi(x)\|_{\mathcal{H}} = \|K_x\|_{\mathcal{H}} \leq A, \forall x \in \mathcal{B}$. In other words, for every probability measure $\mu$ on $\mathcal{B}$, the probability measure $\Phi_{\#}\mu$ is supported on the ball in $\mathcal{H}$ centered at the origin with radius $A$. By Theorem B.1 with $\delta = 1$, we obtain

$$\mathbb{E}\mathcal{KMS}_1(\mu, \widehat{\mu}_n) = \mathbb{E}\mathcal{MS}_1\Big(\Phi_{\#}\mu, \Phi_{\#}\widehat{\mu}_n\Big) \leq \frac{AC}{\sqrt{n}}.$$

Since $\Phi_{\#}\mu$ and $\Phi_{\#}\widehat{\mu}_n$ are supported on the ball of $\mathcal{H}$ centered at the origin with radius $A$, it holds that

$$\mathcal{MS}_p\Big(\Phi_{\#}\mu, \Phi_{\#}\widehat{\mu}_n\Big) \leq \Big[\mathcal{MS}_1\Big(\Phi_{\#}\mu, \Phi_{\#}\widehat{\mu}_n\Big) \cdot (2A)^{p-1}\Big]^{1/p}.$$

In other words,

$$\mathcal{KMS}_p(\mu, \widehat{\mu}_n) \leq \Big[\mathcal{KMS}_1(\mu, \widehat{\mu}_n) \cdot (2A)^{p-1}\Big]^{1/p}. \tag{15}$$

It follows that

$$\begin{aligned}
\mathbb{E}\mathcal{KMS}_p(\mu, \widehat{\mu}_n) &= \mathbb{E}\mathcal{MS}_p\Big(\Phi_{\#}\mu, \Phi_{\#}\widehat{\mu}_n\Big) \\
&\leq \mathbb{E}\Big[\mathcal{MS}_1\Big(\Phi_{\#}\mu, \Phi_{\#}\widehat{\mu}_n\Big) \cdot (2A)^{p-1}\Big]^{1/p} \\
&\leq \Big\{\mathbb{E}\Big[\mathcal{MS}_1\Big(\Phi_{\#}\mu, \Phi_{\#}\widehat{\mu}_n\Big) \cdot (2A)^{p-1}\Big]\Big\}^{1/p} \\
&\leq \Big\{\frac{AC}{\sqrt{n}} \cdot (2A)^{p-1}\Big\}^{1/p} = 2^{1-1/p} AC^{1/p} \cdot n^{-1/(2p)}.
\end{aligned}$$

(II) For the second part, we re-write $\mathcal{KMS}_1(\mu, \widehat{\mu}_n)$ with $\widehat{\mu}_n = \frac{1}{n}\sum_{i=1}^n \delta_{x_i}$ using the Kantorovich dual reformulation of OT:

$$\mathcal{KMS}_1(\mu, \widehat{\mu}_n) = \sup_{\substack{f \in \mathcal{H}: \|f\|_{\mathcal{H}} \leq 1, \\ g \text{ is 1-Lipschitz with } g(0) = 0}} \left| \frac{1}{n}\sum_{i=1}^n \Big(g(f(x)) - \mathbb{E}_{x \sim \mu}[g(f(x))]\Big) \right|,$$

where the additional constraint $g(0) = 0$ does not impact the optimal value of the OT problem. In other words, one can represent

$$\mathcal{KMS}_1(\mu, \widehat{\mu}_n) = \sup_{h \in \mathfrak{H}} \left| \frac{1}{n}\sum_{i=1}^n h(x_i) \right|,$$

where the function class

$$\mathfrak{H} = \Big\{x \mapsto g(f(x)) - \mathbb{E}_{x \sim \mu}[g(f(x))] : g \text{ is 1-Lipschitz with } g(0) = 0, \quad f \in \mathcal{H}, \|f\|_{\mathcal{H}} \leq 1\Big\}.$$

Consequently, for any $x$,

$$|g(f(x))| = |g(f(x)) - g(0)| \leq |f(x)| = |\langle f, K_x \rangle_{\mathcal{H}}| \leq \|f\|_{\mathcal{H}}\|K_x\|_{\mathcal{H}} \leq A.$$

One can apply Theorem B.2 with $\mathcal{F} \equiv \mathfrak{H}$, $a_{i,h} \equiv -A - \mathbb{E}_{x \sim \mu}[g(f(x))]$, $b_{i,h} \equiv A - \mathbb{E}_{x \sim \mu}[g(f(x))]$, where $h(x) = g(f(x)) - \mathbb{E}_{x \sim \mu}[g(f(x))]$, to obtain

$$\mathbb{P}\left\{\mathcal{KMS}_1(\mu, \widehat{\mu}_n) \geq \mathbb{E}\left[\mathcal{KMS}_1(\mu, \widehat{\mu}_n)\right] + \delta\right\} \leq \exp\left(-\frac{n\delta^2}{4(2A)^2}\right) = \exp\left(-\frac{n\delta^2}{16A^2}\right).$$

Or equivalently, the following relation holds with probability at least $1 - \alpha$:

$$\mathcal{KMS}_1(\mu, \widehat{\mu}_n) \leq \mathbb{E}\left[\mathcal{KMS}_1(\mu, \widehat{\mu}_n)\right] + 4An^{-1/2}\sqrt{\log\frac{1}{\alpha}} \leq An^{-1/2}\left(C + 4\sqrt{\log\frac{1}{\alpha}}\right).$$

By the relation (15), we find that with probability at least $1 - \alpha$,

$$\mathcal{KMS}_p(\mu, \widehat{\mu}_n)$$
$$\leq \left[An^{-1/2}\left(C + 4\sqrt{\log\frac{1}{\alpha}}\right) \cdot (2A)^{p-1}\right]^{1/p} = 2^{1-1/p}A\left(C + 4\sqrt{\log\frac{1}{\alpha}}\right)^{1/p} \cdot n^{-1/(2p)}.$$

$$\square$$

$\square$

We now show the proof of Theorem 3.2(II). By the triangle inequality, with probability at least $1 - 2\alpha$, it holds that

$$\mathcal{KMS}_p(\widehat{\mu}_n, \widehat{\nu}_n) \leq \mathcal{KMS}_p(\mu, \widehat{\mu}_n) + \mathcal{KMS}_p(\nu, \widehat{\nu}_n) + \mathcal{KMS}_p(\mu, \nu)$$
$$\leq 2 \cdot 2^{1-1/p}A\left(C + 4\sqrt{\log\frac{1}{\alpha}}\right)^{1/p} \cdot n^{-1/(2p)} + \mathcal{KMS}_p(\mu, \nu)$$
$$\leq 4A\left(C + 4\sqrt{\log\frac{1}{\alpha}}\right)^{1/p} \cdot n^{-1/(2p)} + \mathcal{KMS}_p(\mu, \nu).$$

Then, substituting $\alpha$ with $\alpha/2$ gives the desired result.

*Proof of Corollary 3.3.* It remains to show the type-II risk when proving this corollary. In particular,

$$\text{Type-II Risk} = \mathbb{P}_{H_1}\left\{\mathcal{KMS}_p(\widehat{\mu}_n, \widehat{\nu}_n) < \Delta(n, \alpha)\right\}$$
$$= \mathbb{P}_{H_1}\left\{\mathcal{KMS}_p(\mu, \nu) - \mathcal{KMS}_p(\widehat{\mu}_n, \widehat{\nu}_n) \geq \mathcal{KMS}_p(\mu, \nu) - \Delta(n, \alpha)\right\}$$
$$\leq \mathbb{P}_{H_1}\left\{\left|\mathcal{KMS}_p(\mu, \nu) - \mathcal{KMS}_p(\widehat{\mu}_n, \widehat{\nu}_n)\right| \geq \mathcal{KMS}_p(\mu, \nu) - \Delta(n, \alpha)\right\}$$
$$\leq \frac{\mathbb{E}\left|\mathcal{KMS}_p(\mu, \nu) - \mathcal{KMS}_p(\widehat{\mu}_n, \widehat{\nu}_n)\right|}{\mathcal{KMS}_p(\mu, \nu) - \Delta(n, \alpha)},$$

where the last relation is based on the Markov inequality and the assumption that $\mathcal{KMS}_p(\mu, \nu) - \Delta(n, \alpha) > 0$. Based on the triangular inequality, we can see that

$$\mathbb{E}\left|\mathcal{KMS}_p(\mu, \nu) - \mathcal{KMS}_p(\widehat{\mu}_n, \widehat{\nu}_n)\right| \leq \mathbb{E}[\mathcal{KMS}_p(\mu, \widehat{\mu}_n)] + \mathbb{E}[\mathcal{KMS}_p(\nu, \widehat{\nu}_n)] \leq 2AC^{1/p} \cdot n^{-1/(2p)}.$$

Combining these two upper bounds, we obtain the desired result. $\square$

## C. Sufficient Condition for Positive Definiteness of Matrix $G$

To implement our computational algorithm, one needs to assume the gram matrix

$$G = [K(x^n, x^n), -K(x^n, y^n); -K(y^n, x^n), K(y^n, y^n)]$$

to be strictly positive definite. By the Lemma on the Schur complement (see, e.g., [5, Lemma 4.2.1]), It can be showed that its necessary and sufficient condition should be

$$G' = [K(x^n, x^n), K(x^n, y^n); K(y^n, x^n), K(y^n, y^n)]$$

is strictly positive definite. By Wendland [79], this requires our data points $\{x_1, \ldots, x^n, y_1, \ldots, y_n\}$ are pairwise distinct and $K(x, y)$ is of the form $K(x, y) = \Phi(x - y)$, with $\Phi(\cdot)$ being continuous, bounded, and its Fourier transform is non-negative and non-vanishing. For instance, Gaussian kernel $K(x, y) = e^{-\|x-y\|_2^2/\sigma^2}$ or Bessel kernel $K(x, y) = (c^2 + \|x\|_2^2)^{-\beta}, x \in \mathbb{R}^d, \beta > d/2$ satisfies our requirement.

## D. Reformulation for $2$-KMS Wasserstein Distance in (KMS)

In this section, we derive the reformulation for computing 2-KMS Wasserstein distance:

$$\max_{f \in \mathcal{H}, \|f\|_{\mathcal{H}}^2 \leq 1} \left\{ \min_{\pi \in \Gamma_n} \sum_{i,j \in [n]} \pi_{i,j} |f(x_i) - f(y_j)|^2 \right\}. \tag{16}$$

Based on the expression of $f$ in (7), we reformulate the problem above as

$$\max_{a_x, a_y \in \mathbb{R}^n} \left\{ \min_{\pi \in \Gamma_n} \sum_{i,j \in [n]} \pi_{i,j} \left| \sum_{l \in [n]} a_{x,l} K(x_i, x_l) - \sum_{l \in [n]} a_{y,l} K(y_j, y_l) \right|^2 \right\}, \tag{17a}$$

subject to the constraint

$$\left\| \sum_{i=1}^n a_{x,i} K(\cdot, x_i) - \sum_{i=1}^n a_{y,i} K(\cdot, y_i) \right\|_{\mathcal{H}}^2$$
$$= \left\langle \sum_{i=1}^n a_{x,i} K(\cdot, x_i) - \sum_{i=1}^n a_{y,i} K(\cdot, y_i) \sum_{i=1}^n a_{x,i} K(\cdot, x_i) - \sum_{i=1}^n a_{y,i} K(\cdot, y_i) \right\rangle$$
$$= \sum_{i,j \in [n]} a_{x,i} a_{x,j} \langle K(\cdot, x_i), K(\cdot, x_j) \rangle + \sum_{i,j \in [n]} a_{y,i} a_{y,j} \langle K(\cdot, y_i), K(\cdot, y_j) \rangle - 2 \sum_{i,j \in [n]} a_{x,i} a_{y,j} \langle K(\cdot, x_i), K(\cdot, y_j) \rangle \leq 1. \tag{17b}$$

One can re-write (17) in compact matrix form. If we define

$$s = [a_x; a_y],$$
$$M'_{i,j} = [(K(x_i, x_l) - K(y_i, x_l))_{l \in [n]}; (K(y_j, y_l) - K(x_i, y_l))_{l \in [n]}],$$
$$G = [K(x^n, x^n), -K(x^n, y^n); -K(y^n, x^n), K(y^n, y^n)] \in \mathbb{R}^{2n \times 2n},$$

Problem (17) can be reformualted as

$$\max_{s \in \mathbb{R}^{2n}} \left\{ \min_{\pi \in \Gamma_n} \sum_{i,j \in [n]} \pi_{i,j} \left| s^{\mathrm{T}} M'_{i,j} \right|^2 : \quad s^{\mathrm{T}} G s \leq 1 \right\}. \tag{18}$$

Take Cholesky decomposition $G^{-1} = U U^{\mathrm{T}}$ and use the change of variable approach to take $\omega = U^{-1} s$, Problem (18) can be further reformulated as

$$\max_{\omega \in \mathbb{R}^{2n}} \left\{ \min_{\pi \in \Gamma_n} \sum_{i,j \in [n]} \pi_{i,j} \left( \langle \omega, U^{\mathrm{T}} M'_{i,j} \rangle \right)^2 : \quad \omega^{\mathrm{T}} \omega \leq 1 \right\}. \tag{19}$$

After defining $M_{i,j} = U^{\mathrm{T}} M'_{i,j}$ and observing that the inequality constraint $\omega^{\mathrm{T}} \omega \leq 1$ will become tight, we obtain the desired reformulation as in (9).

## E. Proof of Theorem 4.2

The general procedure of NP-hardness proof is illustrated in the following diagram: Problem (9) contains the **(Fair PCA with rank-1 data)** as a special case, whereas this special problem further contains **(Partition)** (which is known to be NP-complete) as a special case. After building these two reductions, we finish the proof of Theorem 4.2.

Figure 7. Proof outline of Theorem 4.2

**Claim 1.** Problem (9) contains Problem (**Fair PCA with rank-1 data**).

*Proof of Claim 1.* Given vectors $A_1, \ldots, A_n$, we specify

$$M_{1,:} \triangleq \{M_{1,1}, M_{1,2}, \ldots, M_{1,n}\} = \{A_1, \ldots, A_n\},$$

and $M_{i,:} \triangleq \{M_{i,1}, M_{i,2}, \ldots, M_{i,n}\}, i = 2, \ldots, n$ is specified by circularly shifting the elements in $M_{1,:}$ by $i - 1$ positions. For instance, $M_{2,:} = \{A_n, A_1, \ldots, A_{n-1}\}$. For the inner OT problem in (9), it suffices to consider deterministic optimal transport $\pi$, i.e.,

$$\pi_{i,j} = \begin{cases} 1/n, & \text{if } j = \sigma(i), \\ 0, & \text{otherwise} \end{cases}$$

for some bijection mapping $\sigma : [n] \to [n]$. The cost matrix for the inner OT is actually a circulant matrix:

$$\left((M_{i,j}^{\mathrm{T}}\omega)^2\right)_{i,j} = \begin{pmatrix} (A_1\omega)^2 & (A_2\omega)^2 & \cdots & (A_n\omega)^2 \\ (A_n\omega)^2 & (A_1\omega)^2 & \cdots & (A_{n-1}\omega)^2 \\ \vdots & \vdots & \ddots & \vdots \\ (A_2\omega)^2 & (A_3\omega)^2 & \cdots & (A_1\omega)^2 \end{pmatrix}.$$

When considering the feasible circularly shifting bijection mapping (e.g., $\sigma(i) = (i + j) \mod n, \forall i \in [n]$ for $j = 0, 1, \ldots, n - 1$), we obtain the upper bound on the optimal value of the inner OT problem in (9):

$$\min_{\pi \in \Gamma_n} \sum_{i,j} \pi_{i,j}(M_{i,j}^{\mathrm{T}}\omega)^2 \le \min_{i \in [n]} (A_i^{\mathrm{T}}\omega)^2 = \min_{i \in [n]} \omega^{\mathrm{T}} A_i A_i^{\mathrm{T}} \omega.$$

On the other hand, for any bijection mapping $\sigma$, the objective of the inner OT problem in (9) can be written as a convex combination of $(A_1^{\mathrm{T}}\omega)^2, \ldots, (A_n^{\mathrm{T}}\omega)^2$, and thus,

$$\min_{\pi \in \Gamma_n} \sum_{i,j} \pi_{i,j}(M_{i,j}^{\mathrm{T}}\omega)^2 \ge \min_{\alpha \in \mathbb{R}_n^+, \sum_i \alpha_i = 1} \left\{ \sum_i \alpha_i (A_i^{\mathrm{T}}\omega)^2 \right\} \ge \min_{i \in [n]} (A_i^{\mathrm{T}}\omega)^2.$$

Since the upper and lower bounds match with each other, we obtain

$$\min_{\pi \in \Gamma_n} \sum_{i,j} \pi_{i,j}(M_{i,j}^{\mathrm{T}}\omega)^2 = \min_{i \in [n]} \omega^{\mathrm{T}} A_i A_i^{\mathrm{T}} \omega,$$

and consequently,

$$\max_{\omega: \|\omega\|_2 = 1} \left\{ \min_{\pi \in \Gamma_n} \sum_{i,j} \pi_{i,j}(M_{i,j}^{\mathrm{T}}\omega)^2 \right\} = \max_{\omega: \|\omega\|_2 = 1} \left\{ \min_{i \in [n]} \omega^{\mathrm{T}} A_i A_i^{\mathrm{T}} \omega \right\},$$

which justifies Problem (9) contains Problem (**Fair PCA with rank-1 data**). □

**Claim 2.** Problem (**Fair PCA with rank-1 data**) contains Problem (**Partition**).

It is noteworthy that Claim 2 has previously been proved by [68]. For the sake of completeness, we provide the proof here.

*Proof of Claim 2.* Consider the norm minimization problem

$$P = \min_{\omega} \left\{ \|\omega\|_2^2 : \min_{i \in [n]} (\omega^{\mathrm{T}} A_i)^2 \geq 1 \right\}. \tag{20}$$

and the scaled problem

$$\max_{\omega} \left\{ \min_{i \in [n]} (\omega^{\mathrm{T}} A_i)^2 : \|\omega\|_2^2 = P \right\}. \tag{21}$$

We can show that Problem (**Fair PCA with rank-1 data**) is equivalent to (21), whereas (21) is equivalent to (20). Indeed,

- For the first argument, for any optimal solution from Problem (**Fair PCA with rank-1 data**), denoted as $\omega^*$, one can do the scaling to consider $\tilde{\omega}^* = \sqrt{P}\omega$, which is also optimal to (21), and vise versa.

- For the second argument, let $\omega_1, \omega_2$ be optimal solutions from (20), (21), respectively. Since $P$ is the optimal value of (20), one can check that $\omega_1$ is a feasible solution to (21). Since $\min_{i \in [n]} (\omega_1^{\mathrm{T}} A_i)^2 \geq 1$, by the optimality of $\omega_2$, it holds that $\min_{i \in [n]} (\omega_2^{\mathrm{T}} A_i)^2 \geq 1$, i.e., $\omega_2$ is a feasible solution to (20). Since $\|\omega_2\|_2^2 = P$, $\omega_2$ is an optimal solution to (20). Reversely, one can show $\omega_1$ is an optimal solution to (21): suppose on the contrary that there exists $\bar{\omega}_1$ such that $\|\bar{\omega}_1\|_2^2 = P$ and $\min_{i \in [n]} (\bar{\omega}_1^{\mathrm{T}} A_i)^2 > \min_{i \in [n]} (\omega_1^{\mathrm{T}} A_i)^2 \geq 1$, then one can do a scaling of $\bar{\omega}_1$ such that $\min_{i \in [n]} (\bar{\omega}_1^{\mathrm{T}} A_i)^2 = 1$ whereas $\|\bar{\omega}_1\|_2^2 > P$, which contradicts to the optimality of $P$. Combining both directions, we obtain the equivalence argument.

Thus, it suffices to show (20) contains Problem (**Partition**). Define $a = (a_i)_{i \in [n]}, Q = I_n + aa^{\mathrm{T}}$, and assume $Q$ admits Cholesky factorization $Q = S^{\mathrm{T}}S$. Then we create the vector $A_i = S^{-\top} e_i$, where $e_i$ is the $i$-th unit vector of length $n$. Then, it holds that

$$
\begin{aligned}
&\overset{(20)}{=} \min_{\omega} \left\{ \|\omega\|_2^2 : \min_{i \in [n]} ((S^{-1}\omega)^{\mathrm{T}} e_i)^2 \geq 1 \right\} \\
&= \min_{\omega} \left\{ \|Sx\|_2^2 : \min_{i \in [n]} (x^{\mathrm{T}} e_i)^2 \geq 1 \right\} \\
&= \min_{\omega} \left\{ x^{\mathrm{T}} Q x : x_i^2 \geq 1 \right\} \\
&= \min_{\omega} \left\{ \sum_{i=1}^{n} x_i^2 + \left( \sum_{i=1}^{n} a_i x_i \right)^2 : x_i^2 \geq 1 \right\} \qquad (*)
\end{aligned}
$$

where the second equality is by the change of variable $x = S^{-1}\omega$, the third equality is by the definitions of $S$ and $e_i$, and the last equality is by the definition of $Q$. The solution to Problem (**Partition**) exists if and only if the optimal value to Problem (*) equals $n$. Thus, we finish the proof of Claim 2. □

## F. Algorithm that Finds Near-optimal Solution to Optimal Transport

In this section, we present the algorithm that returns $\epsilon$-optimal solution to the following OT problem:

$$\min_{\pi \in \Gamma_n} \sum_{i,j} \pi_{i,j} c_{i,j}, \tag{22}$$

where $\{c_{i,j}\}_{i,j}$ is the given cost matrix. Define $\|C\|_\infty = \max_{i,j} c_{i,j}$. In other words, we find $\hat{\pi} \in \Gamma_n$ such that

$$\mathtt{optval}(22) \leq \sum_{i,j} \hat{\pi}_{i,j} c_{i,j} \leq \mathtt{optval}(22) + \epsilon.$$

**Entropy-Regularized OT.** The key to the designed algorithm is to consider the entropy regularized OT problem

$$\min_{\pi \in \Gamma_n} \sum_{i,j} \pi_{i,j} c_{i,j} + \eta \sum_{i,j} \pi_{i,j} \log(\pi_{i,j}),$$

whose dual problem is

$$\min_{v \in \mathbb{R}^n} \left\{ G(v) = \frac{1}{n} \sum_{i=1}^n h_i(v) \right\}, \tag{23}$$

where

$$h_i(v) = \eta \log \sum_j \exp\left( \frac{v_j - c_{i,j} - \eta}{\eta} \right) - \frac{1}{n} \sum_j v_j + \eta(1 + \log n).$$

Given the dual variable $v \in \mathbb{R}^n$, one can recover the primal variable $\pi$ using

$$\pi(v) = \frac{\frac{1}{n} \exp\left( \frac{v_j - c_{i,j} - \eta}{\eta} \right)}{\sum_{j' \in [n]} \exp\left( \frac{v_{j'} - c_{i,j'} - \eta}{\eta} \right)}$$

Algorithm 2 essentailly optimizes the dual formulation (23) with light computational speed.

---

**Algorithm 2** Stochastic Gradient-based Algorithm with Katyusha Momentum for solving OT [82]

---

1: **Input:** Accuracy $\epsilon > 0$, $\eta = \frac{\epsilon}{8 \log n}$, $\epsilon' = \frac{\epsilon}{6 \max_{i,j} c_{i,j}}$, maximum outer iteration $T_{\text{out}}$, and maximum inner iteration $T$.
2: Take $(y_0, z_0, \tilde{\lambda}_0, \lambda_0, C_0, D_0) = (0, 0, 0, 0, 0, 0)$
3: **for** $t = 0, \ldots, T_{\text{out}} - 1$ **do**
4:     $\tau_{1,t} = \frac{2}{t+4}$, $\gamma_t = \frac{\eta}{9\tau_{1,t}}$
5:     $u_t = \nabla \phi(\tilde{\lambda}_t)$
6:     **for** $j = 0, \ldots, T - 1$ **do**
7:       $k = j + tT$
8:       $\lambda_{k+1} = \tau_{1,t} z_k + \frac{1}{2}\tilde{\lambda}_t + (\frac{1}{2} - \tau_{1,t})y_k$
9:       Sample $i$ uniformly from $[n]$, and construct

$$H_{k+1} = u_t + \left( \nabla h_i(\lambda_{k+1}) - \nabla h_i(\tilde{\lambda}_t) \right)$$

10:       Update $z_{k+1} = z_k - \gamma_t \cdot H_{k+1}/2$ and $y_{k+1} = \lambda_{k+1} - \eta H_{k+1}/9$
11:     **end for**
12:     Update $\tilde{\lambda}_{t+1} = \frac{1}{T} \sum_{j=1}^T y_{tT+j}$
13:     Sample $\hat{\lambda}_t$ uniformly from $\{\lambda_{tT+1}, \ldots, \lambda_{tT+T}\}$ and take $D_t = D_t + \text{vec}(\pi(\hat{\lambda}_t))/\tau_{1,t}$
14:     $C_t = C_t + 1/\tau_{1,t}$
15:     $\pi_{t+1} = D_t/C_t$
16: **end for**
17: Query Algorithm 3 to Round $\tilde{\pi} := \pi_{T_{\text{out}}}$ to $\hat{\pi}$ such that $\hat{\pi}\mathbf{1}_n = \frac{1}{n}\mathbf{1}_n$ and $\hat{\pi}^{\mathrm{T}}\mathbf{1}_n = \frac{1}{n}\mathbf{1}_n$
18: **Return** $\hat{\pi}$

---

---

**Algorithm 3** Round to $\Gamma_n$ ([1, Algorithm 2])

---

1: **Input:** $\pi \in \mathbb{R}_+^{n \times n}$
2: $X = \text{diag}(x_1, \ldots, x_n)$, with $x_i = \min\{1, \frac{1}{nr_i(\pi)}\}$. Here $r_i(\pi)$ denotes the $i$-th row sum of $\pi$.
3: $\pi' = X\pi$.
4: $Y = \text{diag}(y_1, \ldots, y_n)$, with $y_j = \min\{1, \frac{1}{nc_i(\pi')}\}$. Here $c_j(\pi')$ denotes the $j$-th column sum of $\pi'$.
5: $\pi'' = \pi' Y$.
6: $\mathbf{e}_r = \frac{1}{n}\mathbf{1}_n - r(\pi'')$, $\mathbf{e}_c = \frac{1}{n}\mathbf{1}_n - c(\pi'')$, where

$$r(\pi'') = (r_i(\pi''))_{i \in [n]}, c(\pi'') = (c_j(\pi''))_{j \in [n]}.$$

7: **Return** $\pi'' + \mathbf{e}_r \mathbf{e}_c^{\mathrm{T}}/\|\mathbf{e}_r\|_1$.

---

**Theorem F.1** ([82, Theorem 3]). *Suppose we specify $T_{out} = \mathcal{O}(\frac{\|C\|_\infty \sqrt{\ln n}}{\epsilon}), T = n$, the number of total iterations (including outer and inner iterations) of Algorithm 2 is $\mathcal{O}(\frac{n\|C\|_\infty \sqrt{\ln n}}{\epsilon})$ with per-iteration cost $\mathcal{O}(n)$. Therefore, the number of arithmetic operations of Algorithm 2 for finding $\epsilon$-optimal solution is $\mathcal{O}(\frac{n^2\|C\|_\infty \sqrt{\ln n}}{\epsilon})$*

## G. Proof of Theorem 4.4

To analyze the complexity of Algorithm 1, we first derive the bias and computational cost of the supgradient estimator $v(S)$ in (12).

**Lemma G.1** (Bias and Computational Cost). *The following results hold:* (I) **(Bias)** $v(S)$ *corresponds to the gradient of* $\widehat{F}(S) = \sum_{i,j} \widehat{\pi}_{i,j} \langle M_{i,j}^{\mathrm{T}} M_{i,j}, S \rangle$, *where $\widehat{\pi}$ is defined in (12) and $|F(S) - \widehat{F}(S)| \le \epsilon$;*
(II) **(Cost)** *The cost for computing (12) is $\mathcal{O}\left(C \cdot n^2 \sqrt{\log n} \epsilon^{-1}\right)$, with $\mathcal{O}(\cdot)$ hiding some universal constant.*

Next, we analyze the error of the inexact mirror ascent framework in Algorithm 1.

**Lemma G.2** (Error Analysis of Algorithm 1). *When taking the stepsize $\gamma = \frac{\log(2n)}{C\sqrt{T}}$, the output $\widehat{S}_{1:T}$ from Algorithm 1 satisfies*

$$0 \le F(S^*) - F(\widehat{S}_{1:T}) \le 2\epsilon + 2C\sqrt{\frac{\log(2n)}{T}}.$$

Combining Lemmas G.1 and G.2, we obtain the complexity for solving (SDR).

*Proof of Lemma G.1.* For the first part, it is noteworthy that $v(S)$ is associated with the objective

$$\widehat{F}(S) = \sum_{i,j} \widehat{\pi}_{i,j} \langle M_{i,j}^{\mathrm{T}} M_{i,j}, S \rangle,$$

where $\widehat{\pi}_{i,j}$ is the $\epsilon$-optimal solution to

$$F(S) = \min_{\pi \in \Gamma_n} \sum_{i,j} \pi_{i,j} \langle M_{i,j}^{\mathrm{T}} M_{i,j}, S \rangle.$$

By definition, it holds that

$$0 \le \widehat{F}(S) - F(S) \le \epsilon.$$

The second part follows from Theorem F.1. $\qquad \square$

The proof of Lemma G.2 replies on the following technical result.

**Lemma G.3** ([55]). *Let $\{S_k\}_{k=1}^{T}$ be the updating trajectory of mirror ascent aiming to solve the maximization of $G(S)$ with $S \in \mathcal{S}_{2n}$, i.e.,*

$$S_{k+1} = \arg\max_{S \in \mathcal{S}_{2n}} \gamma\langle v(S_k), S \rangle + V(S, S_k), \quad k = 1, \dots, T-1.$$

*Here $v(S)$ is a supgradient of $G(S)$, and we assume there exists $M_* > 0$ such that*

$$\|v(S)\|_{Tr} := Trace(v(S)) \le M_*, \qquad \forall S \in \mathcal{S}_{2n}.$$

*Let $\widehat{S}_{1:T} = \frac{1}{T} \sum_{k=1}^{T} S_k$, and $S^*$ be a maximizer of $G(S)$. Define the diameter*

$$D_{\mathcal{S}_{2n}}^2 = \max_{S \in \mathcal{S}_{2n}} h(S) - \min_{S \in \mathcal{S}_{2n}} h(S) = \log(2n).$$

*For constant step size*

$$\gamma = \frac{D_{\mathcal{S}_{2n}}^2}{M_* \sqrt{T}} = \frac{\log(2n)}{M_* \sqrt{T}},$$

*it holds that*

$$0 \le G(S^*) - G(\widehat{S}_{1:T}) \le M_* \sqrt{\frac{4\log(2n)}{T}}.$$

*Proof of Lemma G.2.* Let $S^*$ and $\widehat{S}^*$ be maximizers of the objective $F(\cdot)$ and $\widehat{F}(\cdot)$, then we have the following error decomposition:

$$
\begin{aligned}
&F(S^*) - F(\widehat{S}_{1:T}) \\
={}&\left[F(S^*) - \widehat{F}(S^*)\right] + \left[\widehat{F}(S^*) - \widehat{F}(\widehat{S}^*)\right] + \left[\widehat{F}(\widehat{S}^*) - \widehat{F}(\widehat{S}_{1:T})\right] + \left[\widehat{F}(\widehat{S}_{1:T}) - F(\widehat{S}_{1:T})\right] \\
\leq{}&2\epsilon + \left[\widehat{F}(S^*) - \widehat{F}(\widehat{S}^*)\right] + \left[\widehat{F}(\widehat{S}^*) - \widehat{F}(\widehat{S}_{1:T})\right] \\
\leq{}&2\epsilon + \left[\widehat{F}(\widehat{S}^*) - \widehat{F}(\widehat{S}_{1:T})\right],
\end{aligned}
$$

where the first inequality is because $\|F - \widehat{F}\|_\infty \leq \epsilon$ and

$$
\left|\left[F(S^*) - \widehat{F}(S^*)\right]\right| \leq \epsilon, \left|\left[\widehat{F}(\widehat{S}_{1:T}) - F(\widehat{S}_{1:T})\right]\right| \leq \epsilon;
$$

and the second inequality is because $\widehat{F}(\widehat{S}^*) - \widehat{F}(\widehat{S}_{1:T}) \leq 0$. It remains to bound $\left[\widehat{F}(\widehat{S}^*) - \widehat{F}(\widehat{S}_{1:T})\right]$. It is worth noting that

$$
\|v(S)\|_{\mathrm{Tr}} = \sum_{i,j} \pi_{i,j}\|M_{i,j}M_{i,j}^{\mathrm{T}}\|_{\mathrm{Tr}} \leq \sum_{i,j} \pi_{i,j} \cdot C = C.
$$

Therefore, the proof can be finished by querying Lemma G.3 with $M_* = C$ and stepsize $\gamma = \frac{\log(2n)}{C\sqrt{T}}$. $\qquad\square$

*Proof of Theorem 4.4.* The proof can be finished by taking hyper-parameters such that

$$
2\epsilon \leq \frac{\delta}{2}, \qquad 2C\sqrt{\frac{\log(2n)}{T}} \leq \frac{\delta}{2}.
$$

In other words, we take $\epsilon = \frac{\delta}{4}$ and $T = \lceil \frac{16C^2 \log(2n)}{\delta^2} \rceil$. We follow the proof of Lemma G.2 to take stepsize $\gamma = \frac{\log(2n)}{C\sqrt{T}}$. $\qquad\square$

# H. Proof of Theorems 4.5 and 4.7

We rely on the following two technical results when proving Theorem 4.5.

**Theorem H.1** (Birkhoff-von Neumann Theorem [8])**.** *Consider the discrete OT problem*

$$
\min_{\pi \in \Gamma_n} \sum_{i,j} \pi_{i,j}c_{i,j},. \tag{24}
$$

*There exists an optimal solution $\pi$ that has exactly one entry of $1/n$ in each row and each column with all other entries $0$.*

**Theorem H.2** (Rank Bound, Adopted from [44, Theorem 2] and [43, Lemma 1])**.** *Consider the domain set*

$$
\mathcal{D} = \left\{ S \in \mathbb{S}_m^+ : \text{Trace}(S) = 1 \right\}
$$

*and the intersection of $N$ linear inequalities:*

$$
\mathcal{E} = \left\{ S \in \mathbb{R}^{m \times m} : \langle S, A_i \rangle \geq b_i, i \in [N] \right\}.
$$

*Then, any feasible extreme point in $\mathcal{D} \cap \mathcal{E}$ has a rank at most $1 + \lceil \sqrt{2N + 9/4} - 3/2 \rceil$. Such a rank bound can be strengthened by replacing $N$ by the number of binding constraints in $\mathcal{E}$.*

*Proof of Theorem 4.5.* By taking the dual of inner OT problem, we find (SDR) can be reformulated as

$$
\max_{\substack{S \in \mathcal{S}_{2n} \\ f,g \in \mathbb{R}^n}} \left\{ \frac{1}{n} \sum_{i=1}^n (f_i + g_i) : \quad f_i + g_j \leq \langle M_{i,j}M_{i,j}^{\mathrm{T}}, S \rangle, \quad \forall i,j \in [n] \right\}. \tag{25}
$$

Let $S^*$ be the optimal solution of variable $S$ to the optimization problem above. Then for fixed $S^*$, according to Theorem H.1 and complementary slackness of OT, there exists optimal solutions $(f^*, g^*)$ such that only $n$ constraints out of $n^2$ constraints

in (25) are binding. Moreover, an optimal solution to (SDR) can be obtained by finding a feasible solution to the following intersection of constraints:

$$\text{Find } S \in \mathcal{S}_{2n} \bigcap \mathcal{E} \triangleq \left\{ S : f_i^* + g_j^* \leq \langle M_{i,j} M_{i,j}^{\mathrm{T}}, S \rangle, \ \ i, j \in [n] \right\}.$$

By Theorem H.2, any feasible extreme point from $\mathcal{S}_{2n} \cap \mathcal{E}$ has rank at most $1 + \left\lfloor \sqrt{2n + \frac{9}{4}} - \frac{3}{2} \right\rfloor$. Thus, we pick such a feasible extreme point to satisfy the requirement of Theorem 4.5. $\qquad\square$

*Proof of Theorem 4.7.* Recall that

$$(\text{KMS}) = \max_{\substack{S \succeq 0, \text{Trace}(S)=1, \text{rank}(S)=1, \\ f,g \in \mathbb{R}^n}} \left\{ \frac{1}{n} \sum_{i=1}^{n} (f_i + g_i) : \quad f_i + g_j \leq \langle M_{i,j} M_{i,j}^{\mathrm{T}}, S \rangle, \quad \forall i, j \in [n] \right\}$$

and

$$(\text{SDR}) = \max_{\substack{S \succeq 0, \text{Trace}(S)=1 \\ f,g \in \mathbb{R}^n}} \left\{ \frac{1}{n} \sum_{i=1}^{n} (f_i + g_i) : \quad f_i + g_j \leq \langle M_{i,j} M_{i,j}^{\mathrm{T}}, S \rangle, \quad \forall i, j \in [n] \right\}.$$

It is easy to see $\text{Optval}(\text{KMS}) \leq \text{Optval}(\text{SDR})$. On the other hand, let $(\widehat{S}, \widehat{f}, \widehat{g})$ be an optimal solution to (SDR) such that $\text{rank}(\widehat{S}) \leq k \triangleq 1 + \left\lfloor \sqrt{2n + \frac{9}{4}} - \frac{3}{2} \right\rfloor$. Next, take

$$\zeta \sim \mathcal{N}(0, \widehat{S}), \qquad \xi = \frac{\zeta}{\|\zeta\|_2}, \qquad \widetilde{S} = \xi \xi^{\mathrm{T}}.$$

It can be seen that $\widetilde{S} \succeq 0, \text{Trace}(\widetilde{S}) = 1, \text{rank}(\widetilde{S}) = 1$. Then, for any $\varepsilon \in (0, 1]$ and $\mu > 0$, it holds that

$$\Pr\left\{ \langle M_{i,j} M_{i,j}^{\mathrm{T}}, \widetilde{S} \rangle \geq \varepsilon \cdot (\widehat{f}_i + \widehat{g}_j), \quad \forall i, j \in [n] \right\}$$
$$\geq \Pr\left\{ \langle M_{i,j} M_{i,j}^{\mathrm{T}}, \widetilde{S} \rangle \geq \varepsilon \cdot \langle M_{i,j} M_{i,j}^{\mathrm{T}}, \widehat{S} \rangle, \quad \forall i, j \in [n] \right\}$$
$$= \Pr\left\{ \langle M_{i,j} M_{i,j}^{\mathrm{T}}, \widetilde{S} \rangle \geq \varepsilon \cdot \mathbb{E}[\langle M_{i,j} M_{i,j}^{\mathrm{T}}, \zeta \zeta^{\mathrm{T}} \rangle], \quad \forall i, j \in [n] \right\}$$
$$= \Pr\left\{ \langle M_{i,j} M_{i,j}^{\mathrm{T}}, \zeta \zeta^{\mathrm{T}} \rangle \cdot \|\zeta\|_2^{-2} \geq \varepsilon \cdot \mathbb{E}[\langle M_{i,j} M_{i,j}^{\mathrm{T}}, \zeta \zeta^{\mathrm{T}} \rangle], \quad \forall i, j \in [n] \right\}$$
$$\geq \Pr\left\{ \langle M_{i,j} M_{i,j}^{\mathrm{T}}, \zeta \zeta^{\mathrm{T}} \rangle \geq \frac{\varepsilon}{\mu} \cdot \mathbb{E}[\langle M_{i,j} M_{i,j}^{\mathrm{T}}, \zeta \zeta^{\mathrm{T}} \rangle], \quad \forall i, j \in [n], \quad \|\zeta\|_2^{-2} \geq \mu \right\}$$
$$\geq 1 - \sum_{(i,j) \in [n]} \Pr\left\{ \langle M_{i,j} M_{i,j}^{\mathrm{T}}, \zeta \zeta^{\mathrm{T}} \rangle < \frac{\varepsilon}{\mu} \cdot \mathbb{E}[\langle M_{i,j} M_{i,j}^{\mathrm{T}}, \zeta \zeta^{\mathrm{T}} \rangle] \right\} - \Pr\left\{ \|\zeta\|_2 > \frac{1}{\sqrt{\mu}} \right\}$$
$$\geq 1 - n^2 \cdot \sqrt{\frac{\varepsilon}{\mu}} - \mu.$$

As long as we take $\mu = 33/100, \varepsilon = 4 \cdot (33/100)^3 / n^4$, it holds that

$$\Pr\left\{ \langle M_{i,j} M_{i,j}^{\mathrm{T}}, \widetilde{S} \rangle \geq \varepsilon \cdot (\widehat{f}_i + \widehat{g}_j), \quad \forall i, j \in [n] \right\} \geq 1/100.$$

In other words, there exists $\xi$ such that

$$(\varepsilon \cdot \widehat{f}_i) + (\varepsilon \cdot \widehat{g}_j) \leq \langle M_{i,j} M_{i,j}^{\mathrm{T}}, \xi \xi^{\mathrm{T}} \rangle, \quad \forall i, j \in [n]^2, \qquad \|\xi\|_2 = 1.$$

This indicates that $(\xi \xi^{\mathrm{T}}, \varepsilon \cdot \widehat{f}, \varepsilon \cdot \widehat{g})$ is a feasible solution to (KMS), and consequently,

$$\text{Optval}(\text{KMS}) \geq \frac{\varepsilon}{n} \sum_{i=1}^{n} (\varepsilon \cdot \widehat{f}_i + \varepsilon \cdot \widehat{g}_i) = \varepsilon \cdot (\text{KMS}).$$

$\qquad\square$

---

**Algorithm 4** Rank reduction algorithm for (SDR)

---

1: Run Algorithm 1 to obtain $\delta$-optimal solution to (SDR), denoted as $\widehat{S}$.

```
                              // Step 2:  Find n binding constraints
```

2: Run Hungarian algorithm [41] to solve OT (11) with $S \equiv \widehat{S}$, and obtain an optimal assignment $\sigma : [n] \to [n]$ together with dual optimal solutoin $(f^*, g^*)$.

```
          // Step 3-9:  Calibrate low-rank solution using a greedy algorithm
```

3: Initialize $\delta^* = 1$
4: **while** $\delta^* > 0$ **do**
5: Perform eigendecomposition $\widehat{S} = Q\Lambda Q^\mathrm{T}$, where $\Lambda = \mathrm{diag}(\lambda_1, \ldots, \lambda_r)$ with $\mathrm{rank}(\widehat{S}) = r$
6: Find a direction $Y = Q\Delta Q^\mathrm{T}$, where $\Delta \in \mathcal{S}^r$ is some nonzero matrix satisfying

$$\mathrm{Trace}(\Delta) = 0, \langle Q^\mathrm{T} M_{i,\sigma(i)} M_{i,j}^\mathrm{T} Q, \Delta \rangle = 0, \quad \forall i \in [n].$$

7: **If** such $Y$ does not exist, **then** break the loop.
8: Take new solution $\widehat{S}(\delta^*) := \widehat{S} + \delta^* Y$, where

$$\delta^* = \arg\max_{\delta \geq 0} \Big\{ \delta : \lambda_{\min}(\Lambda + \delta\Delta) \geq 0 \Big\}.$$

9: Update $\widehat{S} = \widehat{S}(\delta^*)$
10: **end while**
11: **Return** $\widehat{S}$

---

## I. Rank Reduction Algorithm

In this section, we develop a rank reduction algorithm that, based on the near-optimal solution (denoted as $\widehat{S}$) returned from Algorithm 1, finds an alternative solution of the same (or smaller) optimality gap while satisfying the desired rank bound in Theorem 4.5.

**Step (i): Find $n$ binding constraints.** First, we fix $S \equiv \widehat{S}$ in (14) and finds the optimal solution $(f^*, g^*)$ such that only $n$ constraints out of $n^2$ constraints are binding. It suffices to apply the Huangarian algorithm [41] to solve the following balanced discrete OT problem

$$\max_{f,g \in \mathbb{R}^n} \left\{ \frac{1}{n} \sum_{i=1}^n (f_i + g_i) : \ f_i + g_j \leq c_{i,j} \right\} = \min_{\pi \in \Gamma_n} \left\{ \sum_{i,j=1}^n \pi_{i,j} c_{i,j} \right\}$$

where $c_{i,j} = \langle M_{i,j} M_{i,j}^\mathrm{T}, \widehat{S} \rangle$. The output of the Huangarian algorithm is a *deterministic* optimal transport that moves $n$ probability mass points from the left marginal distribution of $\pi$ to the right, which is denoted as a bijection $\sigma$ that permutes $[n]$ to $[n]$. Thus, these $n$ binding constraints are denoted as

$$f_i^* + g_{\sigma(i)}^* \leq \langle M_{i,j} M_{i,j}^\mathrm{T}, S \rangle, \quad i \in [n].$$

We denote by the intersection of these $n$ constraints as $\mathcal{E}_n$.

**Step (ii): Calibrate low-rank solution using a greedy algorithm.** Second, let us assume $\widehat{S}$ is not an extreme point of $\mathcal{S}_{2n} \cap \mathcal{E}_n$, since otherwise one can terminate the algorithm to output $\widehat{S}$ following the proof of Theorem 4.5. We run the following greedy rank reduction procedure:

(I) We find a direction $Y \neq 0$, along which $\widehat{S}$ remains to be feasible, and the null space of $\widehat{S}$ is non-decreasing.

(II) Then, we move $\widehat{S}$ along the direction $Y$ until its smallest non-zero eigenvalue vanishes. We update $\widehat{S}$ to be such a new boundary point.

(III) We terminate the iteration until no movement is available.

To achieve (I), denote the eigendecomposition of $\widehat{S}$ with $\text{rank}(\widehat{S}) = r$ as

$$\widehat{S} = \begin{pmatrix} Q & 0 \end{pmatrix} \begin{pmatrix} \Lambda & 0 \\ 0 & 0 \end{pmatrix} \begin{pmatrix} Q^{\mathrm{T}} & 0 \end{pmatrix} = Q\Lambda Q^{\mathrm{T}}$$

where $\Lambda = \text{diag}(\lambda_1, \dots, \lambda_r)$ with $\lambda_1 \geq \cdots \geq \lambda_r > 0$ and $Q \in \mathbb{R}^{2n \times r}$. To ensure $\widehat{S} + \delta Y \in \mathcal{S}_{2n} \cap \mathcal{E}_n$ while $\text{Null}(\widehat{S} + \delta Y) \supseteq \text{Null}(\widehat{S})$, for some stepsize $\delta > 0$, it suffices to take

$$Y = \begin{pmatrix} Q & 0 \end{pmatrix} \begin{pmatrix} \Delta & 0 \\ 0 & 0 \end{pmatrix} \begin{pmatrix} Q^{\mathrm{T}} & 0 \end{pmatrix} = Q\Delta Q^{\mathrm{T}},$$

where $\Delta \in \mathcal{S}^r \setminus \{0\}$ is a symmetric matrix satisfying

$$\text{Trace}(\Delta) = 0, \quad \langle M_{i,j} M_{i,j}^{\mathrm{T}}, Q\Delta Q^{\mathrm{T}} \rangle = 0, \quad i \in [n].$$

To achieve (II), it suffices to solve the one-dimensional optimization

$$\delta^* = \arg\max_{\delta \geq 0} \left\{ \delta : \lambda_{\min}(\Lambda + \delta\Delta) \geq 0 \right\}, \tag{26}$$

where $\lambda_{\min}(\cdot)$ denotes the smallest eigenvalue of a given matrix. the optimization above admits closed-form solution update. Let eigenvalues of $\Delta$ be $\lambda_1' \geq \cdots \geq \lambda_r'$. It suffices to solve

$$\delta^* = \arg\max_{\delta \geq 0} \left\{ \delta : \min_{i \in [r]} (\lambda_i + \delta\lambda_i') \geq 0 \right\}.$$

As long as $\lambda_r' \geq 0$, we return $\delta^* = 0$. Otherwise, let $i$ be an index such that $\lambda_i' \geq 0 > \lambda_{i+1}'$. We take $\delta^* = \max_{i < j \leq r} -\frac{\lambda_j}{\lambda_j'}$ as the desired optimal solution.

The overall algorithm is summarized in Algorithm 4. Its performance guarantee is summarized in Propositions J.1, J.2, and Theorem 4.6.

## J. Proof of Theorem 4.6

The proof of this theorem is separated into two parts.

**Proposition J.1.** *The rank of iteration points in Algorithm 4 strictly decreases. Thus, Algorithm 4 is guaranteed to terminate within $2n$ iterations.*

*Proof.* Assume on the contrary that $\text{rank}(\widehat{S}(\delta^*)) = \text{rank}(\widehat{S}) = r$. Since $\widehat{S}(\delta^*) = Q(\Lambda + \delta^*\Delta)Q^{\mathrm{T}}$, the positive eigenvalues of $\widehat{S}(\delta^*)$ are those of the matrix $\Lambda + \delta^*\Delta$. According to the solution structure of (26), this could happen only when $\Lambda + \delta^*\Delta \succ 0$, i.e., either $\delta^* = 0$ or $\Delta \succeq 0$. For the first case, this algorithm terminates. For the second case, since $\text{Trace}(\Delta) = 0, \Delta \in \mathcal{S}^r$, it implies that $\Delta = 0$, which is a contradiction.

Thus, the rank of the iteration point reduces by at least 1 in each iteration. $\qquad\square$

**Proposition J.2.** *Let $S^*$ be the output of Algorithm 4. Then, it holds that*

(I) *$S^*$ is a $\delta$-optimal solution to (SDR).*

(II) *The rank of $S^*$ satisfies*

$$rank(S^*) \leq 1 + \left\lfloor \sqrt{2n + \frac{9}{4}} - \frac{3}{2} \right\rfloor.$$

*Proof.* Recall the solution $\widehat{S}$ obtained from Step 1 of Algorithm 4 satisfies

$$
\begin{aligned}
F(\widehat{S}) &= \min_{\pi \in \Gamma_n} \sum_{i,j} \pi_{i,j} \langle M_{i,j} M_{i,j}^{\mathrm{T}}, \widehat{S} \rangle \\
&= \max_{f,g \in \mathbb{R}^n} \left\{ \frac{1}{n} \sum_{i=1}^{n} (f_i + g_i) : \ f_i + g_j \leq \langle M_{i,j} M_{i,j}^{\mathrm{T}}, \widehat{S} \rangle \right\} \geq \texttt{objval}(SDR) - \delta.
\end{aligned}
$$

Since Step 2 of Algorithm 4 solves the OT problem exactly, we obtain

$$
\frac{1}{n} \sum_{i=1}^{n} (f_i^* + g_i^*) = F(\widehat{S}) \geq \texttt{objval}(SDR) - \delta
$$

Since Step 3-7 always finds feasible solutions to the $n$ binding constraints

$$
f_i^* + g_{\sigma(i)}^* \leq \langle M_{i,j} M_{i,j}^{\mathrm{T}}, S \rangle, \quad i \in [n],
$$

for any iteration points from Step 3-7, denoted as $\widetilde{S}$, the pair $(\widetilde{S}, f^*, g^*)$ is guaranteed to be the $\delta$-optimal solution to (14), a reformulation of (SDR). Hence we finish the proof of Part (I).

For the second part, assume on the contrary that $r = \mathrm{rank}(S^*) \geq 1 + \left\lfloor \sqrt{2n + \frac{9}{4}} - \frac{3}{2} \right\rfloor$. It implies $n + 1 < r(r+1)/2$. Recall that Step 6 of Algorithm 4 essentially solves a linear system with $n + 1$ constraints and $r(r+1)/2$ variables, so a nonzero matrix $\Delta$ is guaranteed to exist. Thus, one can pick a sufficiently small $\delta > 0$ such that $\lambda_{\min}(\Lambda + \delta \Delta) \geq 0$, which contradicts to the termination condition $\delta^* = 0$. Thus, we finish the proof of Part (II). □

Combining both parts, we start to prove Theorem 4.6.

*Proof.* Algorithm 4 satisfies the requirement of Theorem 4.6. For computational complexity, the computational cost of Step 2 of Algorithm 4 is $\mathcal{O}(n^3)$. In each iteration from Step 3-7, the most computationally expansive part is to solve Step 6, which essentially solves a linear system with $n + 1$ constraints and $r(r+1)/2$ variables. The conservative bound $r \leq 2n$. Hence, the worst-case computational cost of Step 6 (which can be achieved using Gaussian elimination) is

$$
\mathcal{O}((n + 1 + r(r+1)/2) \cdot (n+1)^2) = \mathcal{O}(n^4).
$$

Since Algorithm 4 terminates within at most $2n$ iterations, the overall complexity of it is $\mathcal{O}(n^5)$. □

## K. Numerical Implementation Details

### K.1. Setup for Computing KMS Wasserstein Distance

When implementing our mirror ascent algorithm, for small sample size ($n \leq 200$), we use the exact algorithm adopted from `https://pythonot.github.io/` to solve the inner OT; whereas for large sample size, we use the approximation algorithm adopted from `https://github.com/YilingXie27/PDASGD` to solve this subproblem. For the baseline `BCD` approach, we implement it using the code from `github.com/WalterBabyRudin/KPW_Test/tree/main`.

### K.2. Setup for High-dimensional Hypothesis Testing

For baselines `Sinkhorn Div, SW, MS`, we implement them by calling the well-established package `POT` [20]. For the `ME` baseline, we implement it using the code from `https://github.com/wittawatj/interpretable-test`.

Next, we outline how to generate the datasets for two-sample testing experiments in Fig. 4:

(I) The Gaussian covariance shift dataset was generated by taking

$$
\mu = \mathcal{N}(0, I_d), \nu = \mathcal{N}(0, I_d + \rho E),
$$

where the dimension $d = 200$, the sample size $n = 200$, and $E$ is the all-one matrix. We vary the hyper-parameter $\rho$ from 0 to 0.06.

(II) The Gaussian mixture dataset was generated by taking

$$\mu = \frac{1}{2}\mathcal{N}(0_d, I_d) + \frac{1}{2}\mathcal{N}(0.5 \cdot 1_d, I_d), \quad \nu = \frac{1}{2}\mathcal{N}(0_d, I_d + 0.05E) + \frac{1}{2}\mathcal{N}(0.5 \cdot 1_d, I_d + 0.05E),$$

where the dimension $d = 40$, and we vary the sample size $n$ from 100 to 700.

(III) For MNIST or CIFAR10 examples, we take $\mu$ as the uniform distribution subsampled from the target dataset, denoted as $\mu = p_{\text{data}}$. We take $\nu$ as the one having a change of abundance, i.e., $\nu = 0.85 p_{\text{data}} + 0.15 p_{\text{data of label 1}}$, where $p_{\text{data of label 1}}$ corresponds to the distribution of a subset of the class with label 1.

Finally, we outline the procedure for practically using the KMS Wasserstein distance for two-sample testing, based on the train-test split method:

(I) We first do the $50\%$-$50\%$ training-testing data split such that $x^n = x^{\text{Tr}} \cup x^{\text{Te}}$ and $y^n = y^{\text{Tr}} \cup y^{\text{Te}}$.

(II) Then we compute the optimal nonlinear projector $f$ for training data $(x^{\text{Tr}}, y^{\text{Tr}})$. We specify the testing statistic as the Wasserstein distance between projected testing data $(f_{\#}x^{\text{Te}}, f_{\#}y^{\text{Te}})$.

(III) Then we do the permutation bootstrap strategy that shuffles $(x^{\text{Te}}, y^{\text{Te}})$ for many times, e.g., $L = 500$ times. For each time $t$, the permuted samples are $(x^{\text{Te}}_{(t)}, y^{\text{Te}}_{(t)})$, and we obtain the testing statistic as the Wasserstein distance between projected samples (using the estimated projector $f$, denoted as $\mathcal{W}(f_{\#}x^{\text{Te}}_{(t)}, f_{\#}y^{\text{Te}}_{(t)})$.

(IV) Finally, we obtain the threshold as $(1 - \alpha)$-quantile of the testing statistics for permuted samples, where the type-I error $\alpha = 0.05$.

### K.3. Setup for Human Activity Detection

The MSRC-12 Kinect gesture dataset contains sequences of human body movements recorded by 20 sensors collected from 30 users performing 12 different gestures. We pre-process the dataset by extracting 10 different users such that in the first 600 timeframes, they are performing the throwing action, and in the remaining 300 timeframes, they are performing the lifting action.

### K.4. Setup for Generative Modeling

We follow Deshpande et al. [16] to design the optimization algorithm that solves the problem

$$\min_{\theta} \ \mathcal{D}\big(p_{\text{data}}, (f_{\theta})_{\#}p_{\text{noise}}\big),$$

where $p_{\text{data}}$ denotes the empirical distribution of MNIST dataset, $p_{\text{noise}}$ denotes the Gaussian noise, and $(f_{\theta})_{\#}p_{\text{noise}}$ represents the distribution of the fake image dataset. We specify $f_{\theta}$ as a 4-layer feed-forward neural-net with leaky relu activation, and $\theta$ denotes its weight parameters. We train the optimization algorithm in 30 epoches. From Fig. 6, we observe that the KMS Wasserstein distance provides fake images that are more closer to the ground truth compared with the Sliced Wasserstein distance.

## L. Additional Numerical Study

### L.1. Experiment on Significance Level

Here we add the experiment in the following table to show that as long as we take $\alpha = 0.05$, the practical type-I error of KMS Wasserstein test is guaranteed to be controlled within 0.05.

Table 3. Type-I Error for two-sample testing with Gaussian mixture dataset.

| Method | 20 | 40 | 80 | 160 | 180 | 200 |
|---|---|---|---|---|---|---|
| KMS | $0.061 \pm 0.015$ | $0.055 \pm 0.012$ | $0.043 \pm 0.014$ | $0.059 \pm 0.011$ | $0.042 \pm 0.014$ | $0.062 \pm 0.015$ |

## L.2. Rank Reduction Algorithm

Recall that Theorem 4.5 provides the rank bound regarding some optimal solution from SDR. In this part, we compare the rank of the matrix estimated from Algorithm 1 with our theoretical rank bound based on the CIFAR10 dataset. For a given positive semidefinite matrix, we calculate the rank as the number of eigenvalues greater than the tolerance `1e-6`. The numerical performance is summarized in Table 4. In these cases, one can run our rank reduction algorithm to obtain a low-rank solution. Fig. 8 illustrates the procedure by showing the probability mass values associated with the eigenvectors of the estiamted solution $\widehat{S}$. From the plot, we find that our rank reduction algorithm is capable of producing low-rank solutions even though the matrix size is large.

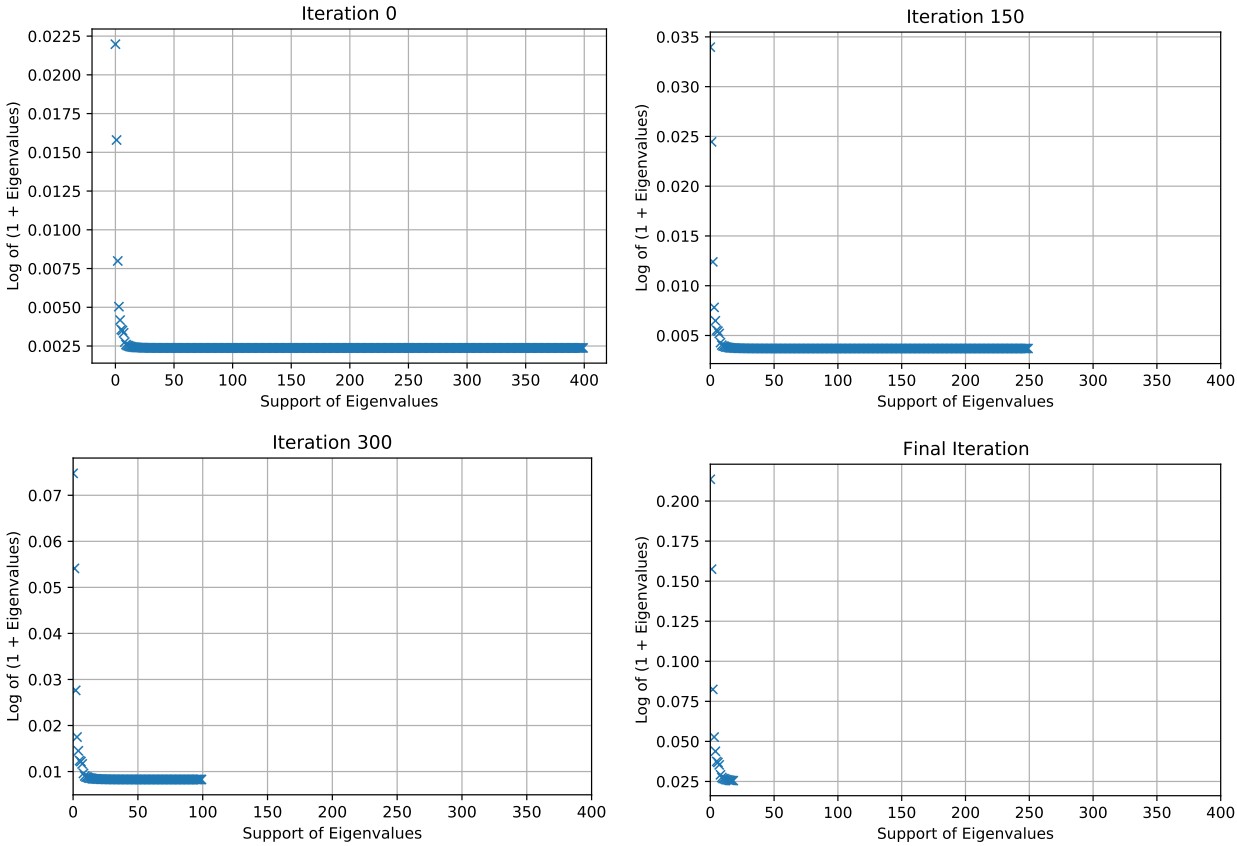

*Figure 8.* Procedure of rank reduction algorithm for CIFAR10 example with sample size $n = 200$.

## L.3. Running Time

*Table 4.* Numerical performance on rank for `CIFAR10` dataset

| Sample Size $n$ | Rank Obtained from Algorithm 1 | Rank Bound from Theorem 4.5 |
|---|---|---|
| 200 | 400 | 19 |
| 250 | 500 | 21 |
| 300 | 600 | 24 |
| 350 | 700 | 26 |
| 400 | 800 | 27 |
| 450 | 900 | 29 |
| 500 | 1000 | 31 |

*Table 5.* Comparison of running time (seconds) different approaches for hypothesis testing, change-detection, and generative modeling experiments.

| | KMS | Sinkhorn Div | MMD | SW | GSW | MS | ME |
|---|---|---|---|---|---|---|---|
| CIFAR-10 Testing ($n = 400$) | 8.21 | 4.07 | 0.017 | 0.91 | 2.58 | 0.064 | 0.17 |
| CIFAR-10 Testing ($n = 500$) | 13.24 | 4.26 | 0.021 | 1.12 | 2.63 | 0.073 | 0.40 |
| CIFAR-10 Testing ($n = 600$) | 18.77 | 4.86 | 0.026 | 1.30 | 2.86 | 0.083 | 0.43 |
| CIFAR-10 Testing ($n = 700$) | 22.13 | 5.81 | 0.033 | 1.56 | 2.94 | 0.096 | 0.41 |
| CIFAR-10 Testing ($n = 800$) | 30.31 | 6.44 | 0.040 | 1.85 | 3.11 | 0.110 | 0.46 |
| CIFAR-10 Testing ($n = 900$) | 34.16 | 7.38 | 0.047 | 2.04 | 3.15 | 0.120 | 0.47 |
| CIFAR-10 Testing ($n = 1000$) | 40.18 | 8.77 | 0.087 | 2.28 | 3.21 | 0.140 | 0.52 |
| Human Activity Detection | 11.16 | 26.30 | 2.42 | 12.88 | 22.48 | 0.58 | 0.91 |
| Generative Modeling | 2530 | 2212 | 744 | 726 | 1716 | 790 | NA |

