# OpenReview forum: "Statistical and Computational Guarantees of Kernel Max-Sliced Wasserstein Distances"
_ICML.cc/2025/Conference — ICML 2025 poster_

### Official Review · Reviewer_LEjS · 2025-03-06

**Overall Recommendation:** 4

**Summary:**

This paper studies statistical and computational properties of Kernel Max-Sliced Wasserstein distances. On the statistical side, the paper's main result is a high-probability bound on the KMS Wasserstein distance between a distribution and the empirical distribution of samples from that distribution. This is in turn used to justify the cutoff for a nonparametric two-sample test, whose power is lower bounded assuming the KMS Wasserstein distance between the true distributions is at least this threshold. On the computational side, although prior work used a representer theorem to reduce computation of the KMS Wasserstein distance between two empirical distributions to a finite-dimensional optimization problem, the present paper shows that this finite-dimensional problem is NP-hard in the worst case. Therefore, the paper proposes a semidefinite relaxation of this problem, which has complexity polynomial in the sample size, precision of the solution, and a certain norm of the Gram matrices of the data. The paper also bounds the rank of the true solution to the semidefinite relaxation and suggests a rank-reduction algorithm to produce solutions with rank near that of the true solution. Finally, the paper provides experimental results demonstrating the computational advantages of the proposed algorithms as well as strong performance on a variety of two-sample testing problems, and applications to change-point detection and generative modeling.

**Claims And Evidence:**

The claims are well justified.

**Essential References Not Discussed:**

N/A

**Experimental Designs Or Analyses:**

The experiments seem extensive, although I did not check the details.

**Methods And Evaluation Criteria:**

Although not the main contribution of the paper, the empirical evaluations effectively demonstrate the utility of the theoretical results and proposed algorithms.

**Other Comments Or Suggestions:**

- Typo: P. 14, lines 761-769: This inequality is missing a $\mathcal{KMS}_p(\mu, \nu)$ term.

- Theorem 2.4 should probably also state that $\mathcal{KMS}_p$ satisfies the triangle inequality, since this is used in the proof of Theorem 3.2, Part II.

- Typo: Line 193, Column 1: "Wassersrein" -> "Wasserstein"

**Other Strengths And Weaknesses:**

- Example 2.5 is quite nice for illustrating the advantages of KMS-Wasserstein distance over MS-Wasserstein distance.
- Generally, the paper is clearly written and easy to read.

**Questions For Authors:**

1) Is it possible to provide a minimax lower bound, or at least an example, showing that the rate $\Delta(n, \alpha)$ in Theorem 3.2 is tight?
2) I don't quite get the motivation for analyzing the rank of the SDR solution. I understand that solutions to the original problem should be rank-1, but what are the ramifications of producing, say, a rank-2 solution vs a rank-3 solution?

**Relation To Broader Scientific Literature:**

1) The relationship/distinctions between the statistical results in the current paper and those in [9] are not very easy to understand from the brief description under Related Work. Can the authors elaborate on what the current statistical results add beyond those of [9]?
2) Corollary 3.3 and Remark 3.4: I think it's worth adding a comment distinguishing dimension-independence of the problem of *estimating KMS Wasserstein distance* from the problem of *two-sample testing using KMS Wasserstein distance*. Under "fair" alternatives (see Ramdas et al. 2015; full reference below), the true KMS Wasserstein distance probably decreases, i.e., the strength of the assumption KMSp(µ, ν) − ∆(n, α) > 0 increases, with dimension. See Ramdas et al. (2015; reference below) for detailed discussion of this phenomenon in the case of MMD.

**References**
Ramdas, A., Reddi, S. J., Póczos, B., Singh, A., & Wasserman, L. (2015, March). On the decreasing power of kernel and distance based nonparametric hypothesis tests in high dimensions. In Proceedings of the AAAI Conference on Artificial Intelligence (Vol. 29, No. 1).

**Theoretical Claims:**

I skimmed the proof of Theorem 3.2, which seemed reasonable.

---

> ### Author Rebuttal · Authors · 2025-04-01
>
> We thank the reviewer's positive comments and provide our response below:
>
> - [relationship and distinctions with literature [9]?] We appreciate the reviewer for highlighting this point. Literature [9] establishes the statistical convergence rate for the empirical Max-Sliced (MS) distance as $O(R \cdot n^{-1/(2p)})$, where $R$ denotes the diameter of the sample space and $n$ is the sample size. This rate is minimax optimal, and the compactness assumption on the sample space is crucial to the analysis. In contrast, our results show that under the bounded kernel assumption (Assumption 1), the empirical KMS distance achieves the same convergence rate of $O(n^{-1/(2p)})$, without requiring the compactness of the original sample space. This allows our result to apply to a broader class of probability distributions. Furthermore, we prove that this rate is also minimax optimal.
> \
> \
> Our statistical analysis builds on the insights from [9]. Specifically, the KMS distance can be interpreted as first mapping the data distributions into an infinite-dimensional Hilbert space via an implicit feature map (induced by the kernel), and then applying the MS distance in that space (see our discussion in Remark 2.6). A key step in our proof leverages the statistical results for the MS distance in Hilbert spaces from [9, Corollary 2.8]. Our bounded kernel assumption ensures that the transformed distributions have bounded support (i.e., finite diameter) in the Hilbert space. We will include this discussion in our revision.
> - [discussion of decreasing power issue of KMS?] We appreciate the reviewer for this insightful comment. Under fair alternative, we believe that the power of our KMS distance will decrease, as the strength of our assumption $\mathrm{KMS}(\mu,\nu) - \Delta(n,\alpha)>0$ increases. We will emphasize this point in our revision and add an additional experiment to illustrate the trend of decreasing power of our method compared to other baselines under such settings.
> - [minimax lower bound?] We provide the following example to demonstrate that the bound $\Delta(n, \alpha)$ is tight. Consider the case where $\mathcal{B} = [0,1]$, the kernel is $k(x,y) = xy$, and the distribution is $\mu = \frac{1}{2}\delta_0 + \frac{1}{2}\delta_1$. Then the empirical KMS distance is given by $\mathrm{KMS}(\mu, \hat{\mu}_n) = \left|\frac{1}{2} - \frac{N}{n}\right|^{-1/p}$, where $N \sim \mathrm{Binom}(n, \frac{1}{2})$. It can be shown that $\mathbb{E}[\mathrm{KMS}(\mu, \hat{\mu}_n)] = \Theta(n^{-1/(2p)})$, thereby confirming that the rate $O(n^{-1/(2p)})$ is optimal in the worst case.
> - [Motivation for rank analysis?] Our motivation for studying the rank of the solution is that it has impact on the quality of the resulting approximation to the original non-convex optimization problem.
> Recall that globally solving the KMS Wassertein distance involves a rank-1 constraint, which our SDR formulation relaxes it.
> When the solution to SDR is of rank higher than $1$, we resort to constructing a feasible rank-1 solution by taking its leading eigenvector. However, if the SDR solution is already low-rank, then the rounding procedure yields a solution that is closer to the ground-truth rank-1 optimum.
> \
> \
> We include an experimental study (see Figure 6 from the Anonymous link https://gofile.io/d/3Pcdxg) that visualizes how our rank reduction algorithm gradually reduces the rank of the SDR solution.
> Originally the optimal solution to SDR is of rank $400$, and our rank reduction algorithm iteratively reduces its rank by $1$ until we obtain the $19$-rank solution. We show the quality of the rounded feasible solution to the original KMS Wasserstein distance problem, and observe the corresponding objective will increase from $0.63$ to $0.68$.
>
>
> - [Typos?] We appreciate the reviewer for pointing out our typos. We will correct them in the revision.

---

### Official Review · Reviewer_Av6C · 2025-03-12

**Overall Recommendation:** 2

**Summary:**

1. **Introduction of the Max-Kernel Sliced Wasserstein Distance (KWS)**
   - The paper presents the Max-Kernel Sliced Wasserstein Distance (KWS), which merges classical max-sliced Optimal Transport (OT) with kernel methods. Data is first mapped to the Reproducing Kernel Hilbert Space (RKHS) through kernel mapping, followed by the computation of max-sliced OT within this space.

2. **Key Properties**
   - The authors demonstrate several properties of KWS:
     - Metric property
     - Sample complexity
     - Existence of solutions

3. **Computation Details**
   - The discussion in the paper is primarily on the computation of a special case of KWS, where the cost function is the squared L2 norm (refer to Equation KMS). The paper notes that this KMS is non-convex (see Eq (9)) and NP-hard (refer to Theorem 4.2). The computational approach is outlined in Algorithm 1, including its complexity.

4. **Applications**
   - KWS is applied in the context of high-dimensional hypothesis testing.

**Claims And Evidence:**

Yes

**Essential References Not Discussed:**

Yes.

The following paper also discusses the OT in RKHS space.

Zhang, Z., Wang, M., & Nehorai, A. (2020). Optimal Transport in Reproducing Kernel Hilbert Spaces: Theory and Applications. *IEEE Transactions on Pattern Analysis and Machine Intelligence*, 42(7). [DOI: 10.1109/TPAMI.2019.2903050](https://doi.org/10.1109/TPAMI.2019.2903050)

**Experimental Designs Or Analyses:**

Yes

**Methods And Evaluation Criteria:**

Yes

**Other Comments Or Suggestions:**

1. **Clarification Needed on Notation**
   - In Equation (9), the notation \(M\) is introduced without explanation. It appears to reference \(M = G - G^T\) from Equation (7). Clarification in the manuscript would be beneficial.

2. **Typographical Corrections**
   - Page 4, line 202, correct the notation from "O(n^1(/2p))" to "O(n^{1/2p})".
   - Page 5, update "\(\pi^*(S) \in \Gamma(S)\)" to "\(\pi^*(S) \in \Gamma_n(S)\)" to ensure accurate mathematical representation.

**Other Strengths And Weaknesses:**

**Weaknesses:**

1. **Advantages of Kernel Method and OT:**
   - The authors are encouraged to further emphasize the benefits of integrating the kernel method with Optimal Transport (OT). Given the paper's proposal of kernel sliced OT (KMS) and its inherent complexities, it is crucial to justify the advantages of this approach over simpler alternatives like applying a non-linear learnable mapping (e.g., MLP) followed by sliced OT. Particularly, the substantial disadvantages of KMS—its NP-hard nature and the expensive, approximative solver—demand a strong argument for using the kernel method instead of just non-linear mapping with classical OT.

2. **Generalized Sliced Wasserstein in Modeling:**
   - Including Generalized Sliced Wasserstein (GSW) in generative modeling experiments would provide a comprehensive evaluation.
   - A comparison of wall-clock times and data sizes across all experiments should be added to assess the practical implementation of the discussed methods.

3. **Time Complexity Concerns:**
   - The paper's stated complexity for solving the inner optimization problem (11) as \(\tilde{\mathcal{O}}(n^2/\epsilon)\) appears to be understated. Standard notation would suggest \(\mathcal{O}(n^2 \ln(n)/\epsilon)\), as supported by recent research [1].
   - The complexity of eigen decomposition required for computing \(h(S)\) is \(O(n^3)\), yet this does not seem to be included in the time complexity discussed in Theorem 4.4.
   - The precision gap (\(\delta\)) addressed in Theorem 4.4 seems to refer to the SDR problem's accuracy rather than the original KMS problem. According to Theorem 4.6, the computational complexity for an approximate solution of KMS is projected at \(n^5\), raising further concerns about its practicality.

4. **Unexplained Superior Performance in Generative Modeling:**
   - There is a lack of explanation for why KMS yields better performance in the generative model experiments. Providing a detailed theoretical justification or analysis would greatly enhance the credibility and acceptance of KMS's claimed effectiveness.

**Reference:**
[1] D. Dvinskikh and D. Tiapkin, "Improved complexity bounds in Wasserstein barycenter problem," in *International Conference on Artificial Intelligence and Statistics*, PMLR, 2021, pp. 1738–1746.

**Questions For Authors:**

See the weakness part.

**Relation To Broader Scientific Literature:**

Classical Optimal Transport (OT) literature often examines sliced optimal transport and generalized sliced optimal transport within the dataset space. This paper introduces an innovative combination by integrating the kernel method with sliced OT, effectively extending sliced OT into the Reproducing Kernel Hilbert Space (RKHS). Thus, the methodology presented fundamentally operates as sliced OT within RKHS.

**Theoretical Claims:**

Yes.

---

> ### Author Rebuttal · Authors · 2025-04-01
>
> We very much appreciate reviewer's insightful comments and now provide our response on a point-by-point basis:
>
> - [OT in RKHS?] Our formulation is closely related to Zhang et al. (2020), which considers Wasserstein distances between pushforward measures $\Phi(\mu)$ and $\Phi(\nu)$ via an implicit kernel map $\Phi$. In Remark 2.6, we show that our KMS distance is equivalent to Max-Sliced Wasserstein in this setting. A key distinction is that our method enjoys sharp statistical convergence rates, which are difficult to obtain in Zhang et al.’s framework due to the curse of dimensionality of Wasserstein.
> Moreover, while their work focuses on kernel embeddings, our motivation differs slightly and is to integrate nonlinear dimensionality reduction with OT. We will clarify this in the revision.
>
> - [OT with a learnable nonlinear map?] We compare our method with $\mathcal{SW}(\Phi(\mu), \Phi(\nu))$ where $\Phi$ is a neural network. Such $\Phi$ is finite-dimensional and sensitive to neural network architectural choices. In contrast, our KMS can also be reformulated as the similar expression (in Remark 2.6) with $\Phi$ being an infinite-dimensional, non-parametric kernel mapping. In addition, our method has theoretical guarantee that $\mathcal{KMS}(\mu,\nu)=0$ iff $\mu=\nu$. This is important for many applications like two-sample testing, whereas it is not guaranteed in fixed neural network mappings.
> \
> \
> While we acknowledge that KMS has higher computational cost, it provides a complementary to neural networks. We will incorporate this discussion into the revision and hope the reviewer sees the merits of our proposed framework.
>
> - [Generative Modeling justification?]
> We revised the setup following [Sliced-Wasserstein Autoencoder] with the new formulation
> $$
> \min_{\phi, \psi} \mathcal{W}(p_{\text{data}}, \psi\circ\phi\circ p_{\text{data}}) + \lambda\cdot\mathcal{D}(q_{\text{prior}}, \phi\circ p_{\text{data}}),
> $$
> where $\psi$ and $\phi$ denote the decoder and encoder, respectively, and $q_{\text{prior}}$ denotes the pre-defined prior distribution on the latent space (uniform distribution on unit circle). We report experiment results in https://gofile.io/d/3Pcdxg (Table 3).  Our KMS Wasserstein distance has competitive performance as indicated by the smallest Fréchet inception distance (FID) score. Also, it has learnt meaningful latent representation, possibly because it utilizes the flexible kernel-projection mapping to compare data distributions, and thereby achieve competitive performance in generative modeling.
> \
> \
> We also clarify that the experiment setup for this part follows from reference [Sliced-Wasserstein Autoencoder]. It could be possible to improve the performance of all baselines by tuning neural network architectures, optimizers, training time, or even random seed. However, our focus in this paper is theoretical and not to develop state-of-the-art generative models. We hope this experiment will be enough to support the empirical value of KMS.
>
> - [Complexity across all experiments?] The runtime for all methods is reported in the anonymous link (Table 2). While KMS is more expensive, the overhead is not prohibitive.
>
> - [Time Complexity Concerns?] We will correct the stated complexity for solving (11) and add the reference as suggested.
>
> - [Complexity of Algorithm 1?] The reviewer is correct. Time complexity of our Algorithm 1 is $\tilde{O}(n^3\delta^{-3})$, not $\tilde{O}(n^2\delta^{-3})$ as previously stated. In our initial submission, we followed established literature on first-order methods (e.g., [Nemirovsky and Yudin, Problem Complexity and Method Efficiency in Optimization]) which analyzes the complexity of constructing supgradient estimators at each iteration. However, we omitted the cost of the proximal gradient projection step induced by $h(S)$ in Eq. (13). This step involves computing the matrix exponential and matrix logarithm, each of which requires $O(n^3)$ time. Fortunately, these operations can be efficiently implemented using well-established software packages.
>
> - [Complexity of solving SDR and rank-reduction?] Solving KMS-Wasserstein is NP-hard; we only analyze its convex relaxation. We also refer to our response to Reviewer LEjS for motivation on the rank-reduction algorithm. We acknowledge that further calibrating the optimal solution from SDR to low-rank space is computationally expansive (whose complexity is $\mathcal{O}(n^5)$), but it is still of theoretical interest if we wish to benefit from the superior performance induced by low-rankness of solution.
>
> - [Typos?] We thank the reviewer and will correct all noted typos and clarify notations in the revision.

---

### Official Review · Reviewer_hSDy · 2025-03-14

**Overall Recommendation:** 3

**Summary:**

This paper establishes statistical theoretical properties of Kernel MSW (K-MSW) distance; the authors provide a finite sample guarantee of K-MSW between empirical probability measures. A performance guarantees are also given when K-MSW is used as a metric of two-sample test. The second part of the paper deals with the computation of 2th power of K-MSW, which is an NP hard problem. The computation relies on a semidefinite relaxation solved with an ineaxact mirroc ascent algorithm.

**Claims And Evidence:**

From a theoretical point view, the authors prove that KMS is a proper metric, and then it has statistical guarantees of finite-sample and two sample as $n$ goes to infinity, the number of support data points. It is worth noting that the convergence rate is dimension-free. KMS is also used as a critical value to determine the rejection of the null hypothesis test between distributions. Concerning the algorithm point, KMS, in a discrete setting, can be cast into a nonconvex max-min optimization problem that is NP-hard (see Theorem 4.2). This problem can be reformulated in a semidefinite relaxation and resolved through an inexact mirror ascent average.

**Essential References Not Discussed:**

The paper discusses the related works, specifically when comparing the performance with many others approaches for testing hypothesis task.

**Experimental Designs Or Analyses:**

The numerical experiments are rich and sound good.

**Methods And Evaluation Criteria:**

The proposed method is evaluated on high dimensional hypothesis testing using synthetic and real datasets. Several baselines approach like Sinkhor divergence, MMD, sliced Wasserstein distance (SW), generalized SWD, max-sliced Wasserstein (MS), and optimized mean-embedding test (ME) are tested on CIFRA-10 dataset. The evaluation metric is the testing power where KMS achieves best performance.

**Other Comments Or Suggestions:**

The paper is well-written and easy to follow.

**Other Strengths And Weaknesses:**

- The paper is well written and easy to follow.
- Proposing statistical guarantees for K-MSW
- Proving that the 2th power of K-MSW is an NP hard problem.
- Solving 2-K-MSW through a semidefinite programming relaxation.

Weaknesses:
KMS suffers from a burden time complexity $\tilde{O}(n^2d^3)$ compared to the vanilla sliced Wasserstein distance that has $\tilde{O}(nd)$. This limits its application to several machine learning pipelines, especially generative modeling.

**Questions For Authors:**

mentioned in weaknesses.

**Relation To Broader Scientific Literature:**

The proposed approach is included in the optimal transport family metrics used for dimension reduction for high-dimensional data.

**Theoretical Claims:**

The proofs sound correct.

---

> ### Author Rebuttal · Authors · 2025-04-01
>
> We very much appreciate reviewer's positive comments and now provide our response on a point-by-point basis:
>
> - [Complexity of KMS?] We would like to clarify that the time complexity of our Algorithm 1 is $\tilde{O}(n^3\delta^{-3})$, not $\tilde{O}(n^2\delta^{-3})$ as previously stated. In our initial submission, we followed established literature on first-order methods (e.g., [Nemirovsky and Yudin, Problem Complexity and Method Efficiency in Optimization]) which analyzes the complexity of constructing supradient estimators at each iteration. However, we omitted the cost of the proximal gradient projection step in Eq. (13). This step involves computing the matrix exponential and matrix logarithm, each of which requires $O(n^3)$ time. These operations are efficiently implemented in modern linear algebra libraries, but their asymptotic cost remains cubic in $n$.
> \
> \
>    Moreover, although Algorithm 1's complexity is independent of the data dimension $d$, it does require precomputing the Gram kernel matrix $G$ (Eq. (8)), and generating the vectors $M_{i,j}'$ as input. This preprocessing step incurs a time complexity of $O(n^3d)$, where the factor of $d$ arises from the kernel computation $k(x,y) = \exp(-\|\|x - y\|\|_2^2 / \sigma^2)$, which scales linearly with $d$ for $x, y \in \mathbb{R}^d$.
> \
> \
> In summary, computing the KMS distance requires time complexity $O(n^3(\delta^{-3} + d))$, which is significantly larger than that of the vanilla sliced Wasserstein distance. We will include a detailed discussion of this point in our revised manuscript.
>
> - [Complexity and Performance Trade-off?] While KMS has in general the highest computational time among the evaluated methods, the overhead is not prohibitive. Importantly, it consistently achieves superior performance, demonstrating that our method is well-suited for practical machine learning applications, albeit with increased computational demands.
> \
> \
> Please also see Table 2 from the Anonymous link https://gofile.io/d/3Pcdxg that reports the numerical running time for hypothesis testing, change-detection, and generative modeling experiments.
> We observe for the change-detection experiment, our approach even has smaller computational time compared with Sinkhorn Divergence, SW, and GSW.  This efficiency arises because the nonlinear projector can be precomputed using pilot data, enabling fast online computation of the test statistic. In contrast, the baseline methods must recompute the statistics at each detection step, resulting in longer runtimes. We will make this point explicit in the revision to highlight the practical utility of our method.
>
> - [Typos?] We appreciate the reviewer for pointing out our typos. We will correct them in the revision.

---

### Decision · Program_Chairs · 2025-05-01

**Decision:**

Accept (poster)

**Comment:**

The focus of the submission is the analysis of the kernel max-sliced (KMS) $p$-Wasserstein distance (Def. 2.3; $1\le p<\infty$). Particularly, the authors
1) prove (sharp) finite-sample guarantee for the KMS $p$-Wasserstein distance between the distribution and its empirical variant (Theorem 3.2),
2) show that the exact computation of the KMS $p$-Wasserstein for $p=2$ is NP-hard (Theorem 4.2), and propose a semidefinite relaxation with complexity (Theorem 4.4) and rank bounds (Theorem 4.5). The efficiency of the proposed approach is demonstrated on both synthetic and real-world experiments including the detection of human activity transition and generative modelling.

Estimating information theoretical measures is a key problem in modern data science with various successful applications. The authors provide tools in this context which have important theoretical and numerical implications, in a well-organized and clearly-written work as it was assessed by the reviewers.